



# Development of the global chemistry-climate coupled model BCC-GEOS-Chem v2.0: improved atmospheric chemistry performance and new capability of chemistry-climate interactions

Ruize Sun[1#], Xiao Lu[2#], Haipeng Lin[3], Tongwen Wu[4,5], Xingpei Ye[1], Lu Shen[1], Xuan Wang[6], Haolin Wang[2], Jingyu Li[2], Ni Lu[1], Jiayin Su[2], Jie Zhang[4,5], Fang Zhang[4,5], Xiaoge Xin[4,5], Xiong Liu[7], Lin Zhang[1*]

[1]Department of Atmospheric and Oceanic Sciences, School of Physics, Peking University, Beijing, China
[2]School of Atmospheric Sciences, Sun Yat-sen University, Zhuhai, Guangdong, China
[3]John A. Paulson School of Engineering and Applied Sciences, Harvard University, Cambridge, MA, United States of America
[4]State Key Laboratory of Disaster Weather Science and Technology, CMA EMPC
[5]CMA Earth System Modeling and Prediction Centre, China Meteorological Administration, Beijing, China
[6]School of Energy and Environment, City University of Hong Kong, Hong Kong SAR, China
[7]Center for Astrophysics | Harvard & Smithsonian, Cambridge, MA 02138, USA

# These authors contributed equally to this work.

Correspondence to: Lin Zhang (zhanglg@pku.edu.cn)

**Abstract.** Interactions between atmospheric chemical compounds and climate have a great impact on the earth system and atmospheric chemistry. However, the online two-way chemistry-climate coupled model, an indispensable tool for quantifying chemistry-climate interactions and projecting future air quality with climate change, remains sparse due to the considerable challenge in model complexity and computational resources. We present the development and evaluation of BCC-GEOS-Chem v2.0, which couples the GEOS-Chem chemical transport model (v14.0.1) with the Beijing Climate Centre Earth System Model (BCC-ESM). Based on the modular framework of BCC-GEOS-Chem v1.0, BCC-GEOS-Chem v2.0 further couples the Harmonized Emissions Component (HEMCO) to manage anthropogenic emission inventories and natural emissions, updates the chemical mechanism, includes the feedback of aerosols and greenhouse gases, and develops the capability for high-resolution simulation. The standard chemical mechanism in the BCC-GEOS-Chem v2.0 features a comprehensive $O_x$-$NO_x$-VOC-halogen-aerosol chemical scheme for the troposphere and the stratosphere. We further evaluate the performance of the BCC-GEOS-Chem v2.0 simulation in representing atmospheric chemistry and compare with the model outputs from the BCC-GEOS-Chem v1.0 and BCC-AGCM-Chem over the simulated time period (2012–2014) at a spatial resolution of T42L26 (approximately 2.8°× 2.8° and 26 vertical layers with a top at 2.914 hPa). BCC-GEOS-Chem v2.0 accurately depicts the primary seasonal and spatial distributions of tropospheric ozone observed by multiple instruments, showing small global mean biases of -2.1−1.8 ppbv for mid-tropospheric (700-400 hPa) ozone concentrations relative to satellite observations, along with high spatial correlation coefficient (*r*) of 0.77−0.92 for individual seasons. It also



demonstrates improved performance in simulating tropospheric carbon monoxide (CO), nitrogen dioxides ($NO_2$), formaldehyde ($CH_2O$) and surface $PM_{2.5}$ compared with both BCC-GEOS-Chem v1.0 and the BCC-AGCM-Chem. The diagnostics of tropospheric ozone budgets (a global tropospheric ozone burden of 355 Tg) and OH concentrations ($0.97 \times 10^6$ molecule $cm^{-3}$) are generally consistent with observation-constrained estimates and multi-model assessment. With the inclusions of aerosol-radiation and aerosol-cloud interactions, BCC-GEOS-Chem v2.0 reproduces the expected impacts of aerosols on radiative and cloud properties, e.g., decreasing shortwave downward solar radiation and outgoing longwave radiation, increasing cloud liquid water, and suppressing precipitations. The high-resolution simulation at T159L72 (approximately 0.75° × 0.75° and 72 vertical layers with a top at 0.01 hPa) further improve the model capability in resolving the fine-scale plume transport dynamics and the pollution hotspot of $NO_2$ and $PM_{2.5}$ as well as the low ozone concentration in high-$NO_x$ environment in wintertime China. The development of the BCC-GEOS-Chem v2.0 model provides a powerful tool to study climate-chemistry interactions and for future projection of global atmospheric chemistry and regional air quality.

## 1 Introduction

Interactions between atmospheric chemical compounds and climate have a profound impact on the earth system and atmospheric chemistry (Isaksen et al., 2009; Fiore et al., 2012). Climate modulates the emissions of chemical compounds from natural and anthropogenic sources, as well as the chemical kinetics, deposition process, and the transport of the advected species (Jacob and Winner, 2009; Abel et al., 2017; Lu et al., 2019a). In turn, changes in greenhouse gases and aerosols alter the radiative budgets of climate system (Haywood and Boucher, 2000; Lohmann and Feichter, 2005; Fan et al., 2016; Alizadeh-Choobari and Gharaylou, 2017). There is an increasing number of climate system models that incorporate interactive atmospheric chemistry from the Phase 5 to the Phase 6 of the Coupled Model Intercomparison Project (CMIP5&CMIP6) (IPCC AR5, 2013, IPCC AR6, 2021). These models play a pivotal role in supporting the Intergovernmental Panel on Climate Change (IPCC) assessment reports, especially in assessing the contribution of both long-lived and short-lived climate forcers (SLCFs) to global change. However, most existing CMIP models employ highly simplified representations of atmospheric chemistry (IPCC AR5, 2013, IPCC AR6, 2021), which introduce substantial uncertainties in quantitative assessments, particularly for SLCFs with high spatiotemporal heterogeneities. Advancing the mechanistic fidelity of atmospheric chemistry in Earth system models is therefore critical for reducing these uncertainties. We previously developed the global atmospheric chemistry general circulation model BCC-GEOS-Chem v1.0 (Lu et al., 2020), a one-way online coupling of Beijing Climate Centre atmospheric general circulation model (BCC-AGCM) and the GEOS-Chem chemical transport model. Here, we present the development of a global chemistry-climate coupled model BCC-GEOS-Chem v2.0, with improved representation of comprehensive troposphere-stratosphere chemistry and new capability to account for radiative-cloud feedbacks from SLCFs, and evaluate its performance in reproducing present-day atmospheric chemistry and feedbacks to climate system.



BCC-GEOS-Chem is designed to integrate GEOS-Chem within BCC-ESM, the Beijing Climate Centre Earth system model developed by the China Meteorological Administration (CMA). BCC-ESM is a fully coupled Earth system model with interactive atmosphere, land, ocean, and sea-ice components. The first BCC-ESM version (BCC-ESM1) utilizes the BCC-AGCM-Chem as the atmospheric chemistry general circulation model, with the chemistry scheme originating from the MOZART2 (Model for OZone And Related chemical Tracers, version 2), a troposphere-only chemical model developed by

the National Center for Atmospheric Research (Horowitz et al., 2003). BCC-ESM1 has participated in a number of research initiatives endorsed by CMIP6, including the Aerosol Chemistry Model Intercomparison Project (AerChemMIP; Collins et al., 2017) and the Coupled Climate–Carbon Cycle Model Intercomparison Project (C4MIP; Jones et al., 2016). GEOS-Chem, initially described by Bey et al. (2001), is a global three-dimensional chemical transport model that includes a detailed ozone–$NO_x$–VOC–aerosol–halogen chemistry scheme. It can both be used offline as a chemical transport model driven by

archived meteorological fields or as an online chemical module driven by weather and climate models (Long et al., 2015; Hu et al., 2018; Lin et al., 2020; Fritz et al., 2022). A defining feature of the GEOS-Chem model lies in its continuous integration of scientific advancements through frequent community-driven updates on model mechanisms and structure (https://github.com/geoschem/geos-chem). The model undergoes rigorous validation through comprehensive benchmarking protocols and continuous evaluation against observational datasets contributed by the international research community (e.g.

Hu et al., 2017; Wang et al., 2022). Its cloud-based architecture ensures open accessibility and computational reproducibility. These characteristics establish GEOS-Chem as a state-of-the-art atmospheric chemistry module in Earth System Models, providing robust scientific capabilities and operational reliability for wide applications on air quality and climate.

BCC-GEOS-Chem v1.0 has enabled the integration of the GEOS-Chem chemical module version 11-02b into the

atmospheric component of BCC-ESM, *i.e.,* the BCC-AGCM (Lu et al., 2020). This is done by separating the chemical module (which simulates all local processes, including convection, chemistry, deposition, and emissions) from the simulation of advective transport in GEOS-Chem, allowing the GEOS-Chem chemical module to operate on one-dimensional vertical columns in a grid-independent manner (Long et al., 2015; Eastham et al., 2018). Transport is done separately in the BCC-AGCM as part of the simulation of atmospheric dynamics. Consequently, the GEOS-Chem chemical

module can be coupled with any ESM grid, where the ESM manages dynamics (transport), allowing adjustable grid types and resolutions. When utilized as an online chemical module in ESMs, GEOS-Chem shares the same code as the offline GEOS-Chem for local processes (convection, chemistry, deposition, and emissions) while using online meteorological fields from the ESMs (Long et al., 2015). This capability ensures that scientific advancements in GEOS-Chem, contributed by the global research community, can be easily incorporated into ESMs, allowing the chemistry of ESM to stay aligned with the

latest version of GEOS-Chem. Validation against suborbital and satellite datasets demonstrates that the BCC-GEOS-Chem v1.0 model successfully reproduces observed spatiotemporal patterns of SLCFs, while also delivering an improved representation of tropospheric oxidation processes. Key metrics such as tropospheric ozone chemical production/loss and the



global hydroxyl radical (OH) budget in BCC-GEOS-Chem v1.0 show enhanced consistency with observational constraints or multi-model assessments, compared with the BCC-AGCM-Chem model implemented in BCC-ESM1 (Lu et al., 2020).


However, there are several limitations in BCC-GEOS-Chem v1.0. Firstly, the GEOS-Chem v11-02b integrated in the BCC-GEOS-Chem v1.0 model employs the TropChem chemical mechanism, which comprises 74 advected tracer species and 91 non-advected species. While this configuration represented state-of-the-art understanding of atmospheric chemical processes at its development stage, subsequent updates of the chemical mechanism and deposition scheme have been introduced to

GEOS-Chem. For example, a more comprehensive halogen chemistry (Wang et al., 2021) and aromatic oxidation mechanism (Bates et al., 2024) has been implemented, demonstrating improved agreement with in situ measurements. The number of chemical species in GEOS-Chem has now expanded to over 200. In particular, the TropChem mechanisms in BCC-GEOS-Chem v1.0 does not include interactive stratospheric chemistry, despite the critical role of stratospheric chemistry in modulating global atmospheric dynamics. Secondly, BCC-GEOS-Chem v1.0 only includes direct radiative

feedback of long-lived greenhouse gases such as carbon dioxide, while radiative-cloud feedbacks associated with chemically active SLCFs such as aerosols are not considered. This key limitation restricts its capability to explore climate-chemistry interactions. Furthermore, BCC-GEOS-Chem v1.0 employs hard-coded emission inputs, requiring manual source code modifications and software recompilation to update emission inventories–a computationally inefficient workflow for iterative experimentation and not adapt to the continuously updated chemical species with rapid updates to the GEOS-Chem

chemical mechanism. These limitations have motivated the development of BCC-GEOS-Chem v2.0, which aims to address these limitations through three key advancements: (1) implementation of state-of-the-art interactive troposphere-stratosphere chemical mechanisms aligned with the latest GEOS-Chem updates, (2) establishment of two-way coupling between atmospheric chemistry and climate to enable interactive feedback processes, and (3) architectural optimization of the codebase to enhance flexibility for file inputs/outputs (including emission inventory) and for future model development.


This paper presents an overview of the BCC-GEOS-Chem v2.0 model, and evaluates the model simulation of present-day atmospheric chemistry and climate with new capability of aerosol feedbacks. In Section 2, we describe the model framework, individual components, and key updates from BCC-GEOS-Chem v1.0. In Section 3, we introduce the observational dataset for model evaluation and the design of model simulation experiment, including a 3-year (2012-2014) base simulation of

BCC-GEOS-Chem v2.0 for comparison with results from the BCC-AGCM-Chem (the chemical module for BCC-ESM1) and BCC-GEOS-Chem v1.0 at the same spatial and temporal resolution, as well as sensitivity simulations for assessing aerosols feedback on the radiation and cloud properties. In Section 4, we evaluate the simulated gaseous pollutants and aerosols with satellite and suborbital observations, and also diagnose the global tropospheric ozone burden and budget. In Section 5, we assess the influences of Aerosol-Radiation-Interactions (ARIs) and Aerosol-Cloud-Interactions (ACIs) on

radiative fluxes and cloud properties. In Section 6, we briefly introduce the capability of BCC-GEOS-Chem v2.0 to perform at high-resolution. Future plans for model development and summary are presented in Section 7.



## 2 Development and description of the BCC-GEOS-Chem v2.0

Figure 1 presents an architectural overview of the BCC-GEOS-Chem v2.0. The previous version BCC-GEOS-Chem v1.0
only enables interactive atmosphere and land modules, with other components such as ocean and sea ice defined as boundary
conditions. The BCC-GEOS-Chem v2.0 has now evolved into a fully coupled Earth system model that integrates
interactively atmospheric dynamic model of BCC-AGCM, land surface model of BCC Atmosphere and Vegetation
Interaction Model (BCC-AVIM), the oceanic component of Modular Ocean Model (MOM), and the sea ice model of SIS
(Section 2.1). These components interact through bidirectional flux exchanges of momentum, energy, water, and carbon
facilitated by the National Centre for Atmospheric Research (NCAR) flux coupler. The BCC-GC-HEMCO interface
(Section 2.2), embedded in BCC-AGCM, is the key component linking the atmospheric module, the Harmonized Emissions
Component (HEMCO) for processing emission inventories and necessary geography data (Keller et al., 2014; Lin et al.,
2021; Section 2.3.1), and the GEOS-Chem chemical module (Section 2.3.2 and 2.3.3). Functional advancements that enable
feedbacks from aerosol and greenhouse gases to climate is introduced separately in Section 2.4.


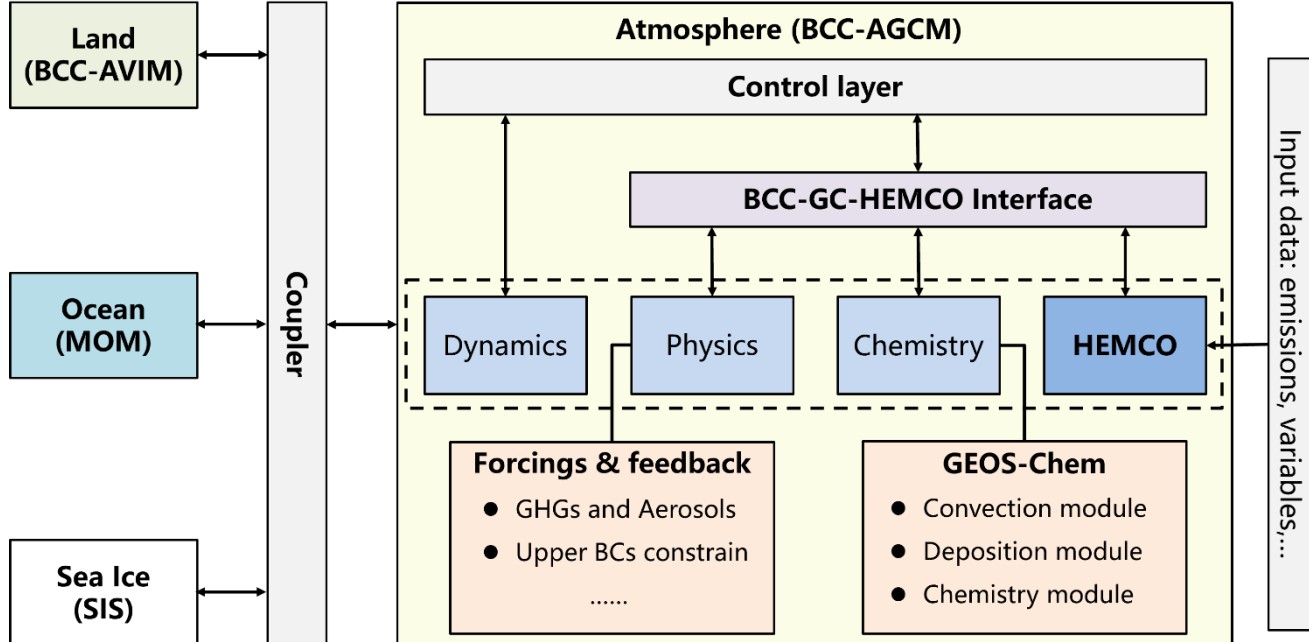

**Figure 1. Architectural overview of the BCC-GEOS-Chem v2.0.**



## 2.1 Atmospheric, Land, Oceanic, and Sea ice components of BCC-GEOS-Chem v2.0

The atmospheric component BCC-AGCM is a global atmospheric spectral model, which can be configured at multiple
horizontal and vertical resolution (e.g., T42L26, T159L72, T382L72, where "T" and affiliated number represents the number
of maximum wavenumbers set by triangular truncation, "L" and affiliated number represents number of vertical levels). In
this study, we use BCC-AGCM version 4.0 at two spectral resolutions, *i.e.*, T42L26 (approximately 2.8° latitude × 2.8°
longitude horizontally and 26 fixed vertical hybrid layers extending from the surface to 2.914 hPa), and T159L72
(approximately 0.75° latitude × 0.75° longitude horizontally and 72 fixed vertical hybrid layers extending from the surface
to 0.01 hPa). We primarily discuss the results of the T42L26 resolution while also comparing them with those from T159L72
in Section 6. The advection of all tracers in BCC-AGCM is handled using a semi-Lagrangian scheme (Williamson and
Rasch, 1989), while vertical diffusion within the boundary layer is parameterized based on the approach of Holtslag and
Boville (1993). Details of physical processes and the dynamical framework of BCC-AGCM are described in Wu et al.
(2019).

BCC-AVIM is an advanced land model with a terrestrial carbon cycle, which includes the biophysical module, plant
physiological module, and soil carbon–nitrogen dynamic module. The biophysical module simulates the exchange of energy,
water, and carbon among the atmosphere, plant canopy, and soil, incorporating 10 soil layers and up to 5 snow layers. The
physiological module performs key vegetation processes, including photosynthesis, respiration, turnover, and mortality, and
diagnoses resulting changes in biomass. The soil carbon-nitrogen dynamic module models biochemical processes, such as
the conversion and decomposition of soil organic carbon. In this study we use BCC-AVIM version 2.2. The detailed
description of BCC-AVIM2 can be found in Li et al. (2019).

The oceanic component MOM employs a tripolar grid with a horizontal resolution of 1° longitude by 1/3° latitude between
30°S and 30°N, while transitioning to 1° longitude by 1° latitude poleward of 60°S and 60°N (Wu et al. 2020). There are 40
z levels in the vertical with 13 levels placed between the surface and a depth of 300 m in the upper ocean. Carbon exchange
between the atmosphere and ocean is computed online in MOM using a biogeochemistry module based on the protocols
established by the Ocean Carbon Cycle Model Intercomparison Project – Phase 2 (OCMIP2). In this study we use MOM
version 5. The sea ice component SIS has the same horizontal resolution as MOM. It has three vertical layers, including one
layer of snow cover and two layers of equally sized sea ice. Additional information about the MOM oceanic component and
SIS sea ice component is available in Wu et al. (2013, 2019).



## 2.2 BCC-GC-HEMCO Interface

The BCC-GC-HEMCO interface (Section 2.2) serves as a central coupler between the atmospheric module, an independent
emission handling module (HEMCO, version 3.6.2, DOI: 10.5281/zenodo.7692950), and the GEOS-Chem chemical module
(version 14.0.1, DOI: 10.5281/zenodo.7271960), as illustrated in Figure 1. It handles the transfer of meteorological variables,
emissions and necessary geography data, and the concentrations of atmospheric constituents between these components. A
key difference in the interface design compared with BCC-GEOS-Chem v1.0 is the implementation of HEMCO as a
standalone emission processing module. At each chemical time step, meteorological variables (Table S2) and atmospheric
constituent concentrations are passed from BCC-AGCM to GEOS-Chem and HEMCO through the BCC-GC-HEMCO
interface. HEMCO returns emission fluxes, diagnosed from read-in offline anthropogenic emission inventory and online
algorithms for specific natural sources using inputted meteorological fields (Section 2.3.1), to BCC-AGCM. The GEOS-
Chem chemistry module is invoked to sequentially perform convection, dry deposition, gas and aerosol chemistry, and wet
deposition. Updated chemical information (e.g. concentration of advected species) is then passed back to BCC-AGCM
through the interface. BCC-AGCM then updates the concentrations of chemical species with input fluxes (emission fluxes
from HEMCO and deposition fluxes diagnosed in GEOS-Chem) through turbulent mixing, and continues to perform
advective transport of the chemical species as part of the simulation of atmospheric dynamics. Minor modifications to
GEOS-Chem source code were made to accommodate the BCC-GC-HEMCO interface, which have been merged to the
GEOS-Chem main code branch (https://github.com/geoschem/geos-chem, last access: 1 April 2025) to ensure future
compatibility between BCC and GEOS-Chem. With the establishment of the BCC-GC-HEMCO interface, in theory, future
replacements of newer versions of GEOS-Chem and HEMCO in the BCC-GEOS-Chem model would only require
downloading and substituting the corresponding source code, without the need for significant modifications to the source
code.

## 2.3 Key processes in BCC-GEOS-Chem v2.0


Here we introduce how emission, chemistry, and dry and wet deposition of atmospheric constituents are processed in BCC-
GEOS-Chem v2.0.

### 2.3.1 Emissions

The Harmonized Emissions Component (HEMCO) was initially developed by Keller et al. (2014) for integrating emission
inventories and algorithms in the GEOS-Chem model. It can provide emissions from a database library of emission
inventories with the capability to select, add, mask, and scale emissions as configured by a text-based configuration file, as
well as a collection of built-in algorithms, referred to as "extensions", which calculate climate-sensitive emissions such as



biogenic, lightning, dust, oceanic sources based on environmental data and model state. The version 3.0 developed by Lin et
al. (2021) further modularized HEMCO enabling coupling to the Community Earth System Model (CESM), while
maintaining a shared core for computing emissions for any other ESM. The enhanced modularity and flexibility of HEMCO
facilitates its seamless integration with the BCC-GEOS-Chem. This architecture also enables efficient incorporation of
community-driven updates to emission inventories or algorithms, without the need for modification and compilation of
model source code in other parts of BCC-GEOS-Chem model. The implementation of HEMCO as a standalone emission
processing module in BCC-GEOS-Chem v2.0 demonstrates a substantial improvement over BCC-GEOS-Chem v1.0. In
BCC-GEOS-Chem v1.0, all emission inventories were directly hard-coded without a unified management framework,
making it difficult to adapt to the continuously updated chemical species in the context of rapidly updates in the GEOS-
Chem chemical mechanism (Lu et al, 2020).

We use HEMCO in BCC-GEOS-Chem v2.0 to manage offline anthropogenic emissions, as well as natural emissions
including biogenic VOCs (Guenther et al., 2012), soil $NO_x$ (Hudman et al., 2012), dust (Zender et al., 2003) and sea salt
(Gong, 2003). These natural emissions can be either calculated online using algorithms that already implemented in HEMCO
driven by meteorological fields provided by BCC-AGCM, or read from pre-archived files using the same algorithms driven
by other meteorological fields (e.g. Weng et al., 2020). Anthropogenic inventories are obtained from the Community
Emissions Data System (CEDS) emission inventory version 2 (McDuffie et al. 2020). Monthly mean biomass burning
emissions are from the Global Emissions Database version 4 (GFED4; Randerson et al., 2018). Calculation of lightning
emissions of NOx are performed in BCC-AGCM rather than in HEMCO for convenience because of its strong link to cloud-
top height (Price and Rind, 1992). Figure S1 shows the spatial distributions of key anthropogenic and natural emissions
averaged over 2012-2014. Table S1 compares the emissions in BCC-GEOS-Chem v2.0 with those for BCC-GEOS-Chem
v1.0 and BCC-AGCM-Chem. Overall, the emission magnitudes and spatial patterns in BCC-GEOS-Chem v2.0 demonstrate
comparable characteristics to both BCC-GEOS-Chem v1.0 and BCC-AGCM-Chem, while also remaining consistent with
values reported in the literature across different models.

### 2.3.2 Atmospheric chemistry

BCC-GEOS-Chem v2.0 uses GEOS-Chem v14.0.1 (https://zenodo.org/records/7271974, The International GEOS-Chem
Community, 2022) as its chemical module. The standard chemical mechanism in GEOS-Chem v14.0.1 features a
comprehensive $O_x$-$NO_x$-VOC-halogen-aerosol chemistry in unified tropospheric–stratospheric chemistry extension (UCX)
scheme (Eastham et al., 2014). The scheme includes 220 advected chemical species and 914 gas-phase chemical reactions
through the Kinetic PreProcessor (KPP) (Damian et al., 2002; Lin et al., 2023). The number of advected tracers is much
higher than that in GEOS-Chem v11-02b used in BCC-GEOS-Chem v1.0, where only 74 chemical species were advected in
troposphere-only chemistry. Updates on chemistry mechanism from GEOS-Chem v11-02b to GEOS-Chem v14.0.1 are



archived at https://github.com/geoschem/geos-chem/releases. Interactive troposphere-stratosphere is thus now available in BCC-GEOS-Chem v2.0, representing as a major improvement compared with BCC-GEOS-Chem v1.0.

The aerosol chemistry of BCC-GEOS-Chem v2.0 includes primary dust, sea salt, primary organic carbon aerosol (POC), primary black carbon aerosol (BC), secondary inorganic aerosols (sulfate, nitrate, ammonium), and secondary organic aerosols (SOAs). Sea salt aerosol masses are simulated in two size ranges: the accumulation mode (dry radii between 0.1 and 0.5 μm) and the coarse mode (dry radii between 0.5 and 4 μm) (Jaeglé et al., 2011). Dust aerosol masses are simulated in four size ranges (radii of 0.1-1.0, 1.0-1.8, 1.8-3.0, and 3.0-6.0 μm; Fairlie et al., 2007). Details of aerosol chemistry and its

interaction with gas-phase chemistry is provided in Lu et al. (2020). Compared with BCC-GEOS-Chem v1.0, key updates of aerosol chemistry include the improved heterogeneous chemistry (e.g. updates in $N_2O_5$, $NO_3$, $NO_2$ reactive uptake, sub-grid cloud $NO_y$ chemistry, aerosol nitrate photolysis) and ISORROPIA Ⅱ (Fountoukis and Nenes, 2007) thermodynamic module. The aerosols feedback on radiation and cloud are implemented in BCC-GEOS-Chem v2.0, as will be introduced in Section 2.4.

### 2.3.3 Dry and Wet Deposition

Dry deposition of gas and aerosols in BCC-GEOS-Chem v2.0 are conducted using the GEOS-Chem standard code, following BCC-GEOS-Chem v1.0. The model estimates deposition velocity following the resistance-in-series scheme (Wesely, 1989), and also includes gravitational settling of particles as described in Zhang et al. (2001). The scheme requires input of geography data such as land types and leaf area index (LAI) to yield surface roughness and canopy resistance. Here

we provide two options in BCC-GEOS-Chem v2.0.  Firstly, we can obtain archived land types (Olson et al., 2001) and the LAI data for each land type from Moderate Resolution Imaging Spectroradiometer (MODIS) used in GEOS-Chem v14.0.1. Alternatively, we can obtain the land types and LAI information from the BCC-AVIM through the coupler, and then computes dry deposition through the GEOS-Chem routines, as done in BCC-GEOS-Chem v1.0. The latter option requires reconciliation of land types used in GEOS-Chem with those used in BCC-AVIM (Lu et al., 2020). We use the first option as

a default option in BCC-GEOS-Chem v2.0, because GEOS-Chem's archived land types contain more detailed vegetation classifications (73 categories) than those in BCC-ACIM (22 categories), ensuring better compatibility with the dry deposition parameterization (Wesely, 1989). The key advantage of the second option is that the land surface parameters used for dry deposition align with those generated by the land surface model, thereby improving its utility for studying atmosphere-land surface interactions. However, uncertainties inherent to the land surface model simulations may propagate

into atmospheric chemistry calculations.

Wet deposition of aerosols and soluble gases by precipitation is calculated by the wet deposition module in GEOS-Chem, following BCC-GEOS-Chem v1.0, which accounts for scavenging in convective updrafts, in-cloud rainout, and below-cloud washout (Liu et al., 2001). In particular, the convective transport of chemical tracers and scavenging in the updrafts is



performed using the GEOS-Chem convection scheme, driven by convection variables diagnosed from BCC-AGCM, instead of using convection module in BCC-AGCM-Chem described by Wu et al. (2020). This is because the BCC-AGCM-Chem scheme for wet deposition is hard-coded and is incompatibility with the updated chemical species. In addition, it lacks the description of scavenging water-soluble species in convective updrafts, which may cause high bias in aerosol concentrations in the upper troposphere (Lin et al., 2024).


### 2.4 The feedback of aerosols and greenhouse gases

One significant update in BCC-GEOS-Chem v2.0, relative to BCC-GEOS-Chem v1.0, is the incorporation of the direct (radiation) and indirect (cloud) climate effects of SLCFs. The incorporation takes advantage of the existing operation of radiative transfer in BCC-AGCM-Chem implemented in BCC-ESM1 (Wu et al., 2020). The longwave radiative transfer

scheme is based on an absorptivity/emissivity formulation detailed by Ramanathan and Downey (1986), incorporating explicit parameterization for major absorbers (Collins et al., 2002a; Kiehl and Briegleb, 1991; Ramanathan and Dickinson, 1979). The shortwave radiation scheme parameterizes gaseous absorption over 19 discrete spectral and pseudo-spectral intervals, as well as the scattering and absorption by cloud droplets and aerosols (Briegleb et al., 1992; Collins et al., 1998). Among the gaseous absorbers, mixing ratios of long-lived greenhouse gases ($CO_2$, $N_2O$, CFC11, and CFC12) and methane

are prescribed and updated by CMIP6 historical forcing data (Section 3), while $H_2O$ and ozone are diagnosed in BCC-GEOS-Chem v2.0.

The aerosol direct effect is calculated based on the mass mixing ratios of bulk aerosols, which are prognostic variables in GEOS-Chem chemical module. Derivation of aerosol single-scattering properties follow a lookup table approach following

Community Atmosphere Model (CAM; Collins et al., 2004). Based on the Optical Properties of Aerosols and Clouds (OPAC) dataset, optical properties of sea salt, OC, and BC are hypothesized to be identical to those of soot and water-soluble aerosols (Hess et al., 1998). Following Wang et al. (2008b), sulfate and nitrate particles are assumed to share the optical properties of ammonium sulfate, while dust optical properties are calculated using Mie theory for each size bin (Zender et al., 2003). Finally, aerosols across different size bins are assumed to be externally mixing, which are subsequently used in radiative

transfer calculation.

The indirect effects of aerosols involve their role in serving as cloud condensation nuclei and exerting influence on cloud properties and precipitation (Rosenfeld et al., 2008). The diagnostic cloud droplet number concentration $N_d$ is calculated using the empirical function Eq. (1) suggested by Quaas et al. (2006):

$$N_d = \exp\left(a_0 + a_1 \ln m_{aer}\right), \tag{1}$$



where $m_{aer}$ is the total mass fraction of all hydrophilic aerosols diagnosed from GEOS-Chem chemical module. $a_0$ and $a_1$ are parameters ($a_0 = 5.1$ and $a_1 = 0.41$) (Boucher and Lohmann, 1995).

The cloud droplet effective radius $R_e$ is estimated by Eq. (2) according to Peng and Lohmann (2003):

$$R_e = \beta R_v = \beta \sqrt[3]{\frac{3LWC}{4\pi \rho_w N_d}}, \qquad (2)$$

where $\beta$ is a scaling factor to obtain $R_e$ from the volume mean radius $R_v$. $\rho_w$ is the water density, $LWC$ is the cloud liquid water content, both are diagnosed by BCC-AGCM.

The aerosol-activated cloud droplets then exert impacts on cloud properties and precipitation, which is taken into account in the parameterization of cloud microphysics processes. The detailed treatment of aerosols feedback on precipitation can be found in Wu et al (2020). The current approach for describing aerosol feedback using a bulk-mass representation of aerosols

in BCC-GEOS-Chem v2.0 is similar to that used in the majority of CMIP5 models, with only two of which include online size-resolved aerosol microphysics (Kodros and Pierce, 2017). This bulk-mass representation of aerosols approach is computationally efficient, but in the meantime do not consider how aerosol microphysics (nucleation, condensation, and coagulation) shape the size distribution. For future model development, we hope to explicitly describe aerosol size distributions by integrating size-resolved schemes such as the Advanced Particle Microphysics (APM) scheme (Yu and Luo,

2009) and the TwO-Moment Aerosol Sectional (TOMAS) microphysics packages (Kodros and Pierce, 2017) from GEOS-Chem to more accurately simulated size-dependent aerosol chemistry and microphysics. Such development will be available through BCC-GC-HEMCO interface once these two schemes become compatible with the GEOS-Chem column structure.

### 3 Model simulation setup and observations for model evaluation

### 3.1 Setup of model experiments

We perform BCC-GEOS-Chem v2.0 simulations and compare the modelled concentrations of key SLCFs against observations and outputs from BCC-GEOS-Chem v1.0 (Lu et al., 2020) and BCC-AGCM-Chem (used in BCC-ESM1, Wu et al. (2020)). These three models represent three generations of development in the atmospheric chemistry module of the BCC Earth System Model. The simulations are performed from 2011 to 2014, with the first year as model spin-up following previous studies (Lu et al. 2020; Schwantes et al., 2022; Fritz et al., 2022). We selected this simulation period because the

output of BCC-GEOS-Chem v1.0 and BCC-AGCM-Chem are available for the same time frame and resolution from Lu et al. (2020) and Wu et al. (2020), respectively.

The initial conditions for atmospheric dynamics and physics in 2011 are obtained from the historical simulations (1850-2014) conducted by BCC-ESM1 under CMIP6 framework, and initial states of chemical tracers are obtained from a GEOS-Chem



simulation over 1995-2017 (Wang et al., 2022). Model results in 2012-2014 are used for evaluation and comparison. External forcing data, including historical concentrations of long-lived greenhouse gases ($CO_2$, $N_2O$, CFCs) and methane, land use forcing, and solar forcing are obtained from the Earth System Grid Federation (ESGF). We note that as BCC-GEOS-Chem only extends to the height of 2.914 hPa, ozone concentrations at the top two layers are set to prescribed monthly climatological values from CMIP6 data package as upper boundary conditions. BCC-AGCM-Chem conducted the

same treatment but included additional stratospheric species ($CH_4$, $N_2O$, NO, $NO_2$, $HNO_3$, CO, and $N_2O_5$), and their concentrations from below the top two layers to the tropopause are relaxed at a relaxation time of 10 d towards the climatology. We also note that the differences of chemical constituents from the three models are not solely due to the chemical mechanisms, but may also arise from variations in the emission inventory, treatment of physical and dynamical processes, inclusion of radiative and cloud feedback, and other factors (e.g. interactions with the ocean model) (Table 1 and

Table S1).

**Table 1.** Summary of the model features and configuration for BCC-GEOS-Chem v2.0, BCC-GEOS-Chem v1.0, and BCC-AGCM-Chem.

|  | BCC-GEOS-Chem v2.0 | BCC-GEOS-Chem v1.0 | BCC-AGCM-Chem |
|---|---|---|---|
| Horizontal resolution | T42 (~2.8° × 2.8°) | | |
| Vertical levels | L26 (up to 2.914 hPa) [a] | | |
| High resolution capability | Available (T159L72) [b] | Unavailable | Unavailable |
| Chemistry | GEOS-Chem v14.0.1 | GEOS-Chem v11-02b | MOZART2 |
|    Advected species | 220 | 78 | 79 |
|    Chemical reactions | 914 | 426 | 168 |
| Emissions | HEMCO module | Hard-coded | Hard-coded |
| Radiative and cloud feedback | Available | Unavailable | Available |
| Interactive ocean and sea ice | Available | Unavailable | Available |
| Original model references | This work | Lu et al. (2020) | Wu et al. (2020) |

[a] Approximately 8 levels in the troposphere and 18 levels in the stratosphere.

[b] The capability for BCC-GEOS-Chem v2.0 to be performed at high resolution has been developed. Results will be discussed in Section 6.

In addition, we perform sensitivity simulations with BCC-GEOS-Chem v2.0 to evaluate the feedback of aerosols on cloud properties and radiation (Table 2). The Base simulation represents the standard BCC-GEOS-Chem v2.0 simulation described above, which considers full aerosol-cloud-radiation interactions. Aerosol direct effect is simulated by linking aerosol optical

properties (such as aerosol optical depth (AOD), single scattering albedo, and asymmetry factor) to the radiation scheme (Chapman et al 2009). The noARIs simulation inactivates the feedback of aerosol on radiation calculation. Aerosol indirect effect is simulated by linking prognostic aerosols to cloud condensation nuclei (CCN). In the noACIs simulation, we use a



prescribed cloud droplet number concentration of 5 cm$^{-3}$ to exclude aerosol indirect effect, following previous studies (Zhao et al., 2017; Zhang et al., 2018; Zhou et al., 2019; Sun et al., 2024). The noARIs and noACIs simulations are initialized in
year 2012, driven by the initial fields archived from the Base model, and model results in 2013 are compared with the Base simulation.

**Table 2.** Setup of sensitivity simulations compared with base simulation in BCC-GEOS-Chem v2.0

| Experiment [a] | Base | noARIs [b] | noACIs [c] |
|---|---|---|---|
| Aerosol direct effect | on | off | on |
| Aerosol indirect effect | on | on | off |

[a] The simulations are performed from 2012 to 2013, with the first year (2012) as model spin-up. [b] The noARIs simulation inactivates the feedback of aerosol
on radiation calculation. [c] The noACIs simulation uses a prescribed cloud droplet number concentration of 5 cm$^{-3}$ to exclude aerosol indirect effect.

### 3.2 Observations and reanalysis data used for model evaluation

We use ozonesonde, surface, and satellite observations, as well as the reanalysis data (Table S3) to evaluate the performance of the BCC-GEOS-Chem v2.0 simulation in representing atmospheric chemistry and climate parameters, with a focus on SLCFs. Ozonesonde measurements are sourced from the World Ozone and Ultraviolet Radiation Data Centre (WOUDC),
operated by the Meteorological Service of Canada within Environment and Climate Change Canada. Ozone from the surface to the stratosphere is measured by balloon-borne ozone electrochemical concentration cell instruments (Tarasick et al., 2019), with the sampling frequencies of 2-4 profiles per week. In this study, we use ozonesonde sites that meet the criteria outlined by Wang et al. (2022). In total, 16 sites across the globe are selected for model evaluation. We then categorize the WOUDC sites into 6 regions (Figure S2a).


Surface ozone observations are collected from individual national monitoring networks, covering major developed and developing regions in the northern hemisphere, including China, the United States of America (USA), Europe, South Korea, and Japan (Figure S2b). We applied data quality control measures to exclude unreliable data following Wang et al. (2024). Surface fine particulate matter (PM$_{2.5}$) data is derived from Shen et al. (2024), which estimates ground-level PM$_{2.5}$ for 2000-
2019 by combining Aerosol Optical Depth (AOD) retrievals from multiple satellite products with the GEOS-Chem chemical transport model and a residual Convolutional Neural Network (CNN) technique.

Additionally, we use the OMI ozone profile (PROFOZ), which consists of 24 vertical layers extending from the surface to 60 km, as retrieved by Liu et al. (2010). This product has been validated through comprehensive comparisons with ozonesonde
data (Huang et al., 2017) and other satellite products (Huang et al., 2018). We also incorporate the tropospheric nitrogen dioxide (NO$_2$) columns (Lamsal et al., 2021) and formaldehyde (CH$_2$O) column (De Smedt et al., 2015), with horizontal resolution of 0.25° × 0.25°. Other satellite observational datasets include carbon monoxide (CO) observations obtained from



the Terra Measurement of Pollution in the Troposphere (MOPITT) satellite instruments (Deeter et al., 2021), and AOD at 550 nm from the Moderate Resolution Imaging Spectroradiometer (MODIS).


We also apply observations and reanalysis data to evaluate the performance of BCC-GEOS-Chem v2.0 on climate simulation and the aerosol's effect. The global monthly observations of radiative properties are obtained from Earth's Radiant Energy System (CERES) sensors onboard the Terra satellite (Kato et al., 2018), including shortwave and longwave cloud radiative forcings (SWCRF and LWCRF), downward shortwave (SWDOWN) radiation, and outgoing longwave

radiation (OLR). The total precipitation is obtained from Global Precipitation Climatology Product (GPCP) satellite data (Huffman et al., 2001). The datasets for evaluation of surface temperature (TS), cloud fraction, and mass fraction of cloud liquid water are from the second version of Modern-Era Retrospective Analysis for Research and Applications (MERRA-2).

## 4 Evaluation and intercomparison of modelled concentrations of short-lived climate forcers

### 4.1 Ozone

We present the spatial and seasonal distributions of global mid-tropospheric ozone derived from OMI satellite observations averaged over 2012–2014, and compare the discrepancies between these observations and the results from models in Figure 2. We analyse ozone at 700–400 hPa where OMI satellite has the peak sensitivity to ozone concentration (Liu et al., 2010). To ensure a proper comparison with observations, OMI averaging kernels and a priori profiles are applied to all model results following Zhang et al. (2010).


Satellite observations indicate high mid-tropospheric ozone levels over the northern midlatitudes both in boreal spring (March-April-May, MAM) and summer (June-July-August, JJA), due to active stratosphere-troposphere exchange (STE) and high photochemical production (Cooper et al., 2014). Elevated ozone levels also are witnessed over southern Africa in boreal autumn (September-October-November, SON), driven by intense biomass burning emissions (Sauvage et al., 2007;

Figure S1). BCC-GEOS-Chem v2.0 captures well the spatial and seasonal distributions of tropospheric ozone observed from OMI satellite, with high spatial correlation coefficients of $r$=0.77-0.92 for different seasons (Figure S3). Compared with satellite observations, BCC-GEOS-Chem v2.0 exhibits overall small global mean biases of mid-tropospheric ozone concentrations, ranging from -2.1 to 1.8 ppbv. Compared with BCC-GEOS-Chem v1.0, BCC-GEOS-Chem v2.0 slightly reduce the ozone low-bias in the northern midlatitudes. However, the model tends to overestimate ozone in the mid-latitudes

especially in the Tibetan Plateau by 4-20 ppbv in boreal spring, which may be attributed to excessive STE as will be demonstrated in Section 4.3.





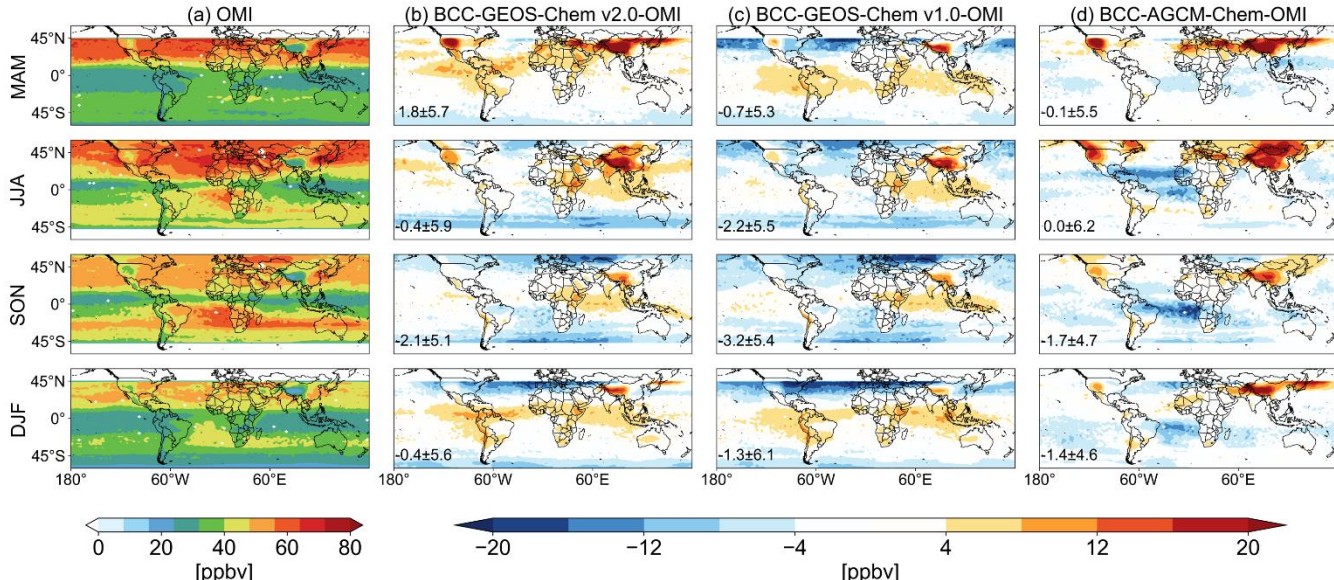

**Figure 2. Spatial and seasonal distributions of global mid-tropospheric ozone averaged for 700-400hPa over 2012–2014 from (a) OMI satellite observation, (b) differences between BCC-GEOS-Chem v2.0 and OMI, (c) differences between BCC-GEOS-Chem v1.0 and OMI, and (d) differences between BCC-AGCM-Chem and OMI. All model outputs are applied with OMI averaging kernels for a proper comparison with the observations.**

Figure 3 presents a comparative analysis of annual mean ozone vertical profiles simulated by BCC-GEOS-Chem v2.0, BCC-GEOS-Chem v1.0, and BCC-AGCM-Chem against WOUDC ozonesonde observations across six regions, averaged over the 2012-2014 period. The results demonstrate that both BCC-GEOS-Chem v2.0 and v1.0 effectively replicate observed ozone concentrations in the lower and middle troposphere (surface to 400 hPa), showing biases generally within 5 ppbv across most regions, except for an enhanced positive bias reaching 15 ppbv in tropical areas. In contrast, BCC-AGCM-Chem exhibits substantially larger positive biases of 8-20 ppbv throughout the lower and middle troposphere. This may be attributed to the better performance of BCC-GEOS-Chem in simulating the major precursors of tropospheric ozone, as will be demonstrated in Section 4.2.

Divergent model performance emerges in the upper troposphere and stratosphere. Both BCC-GEOS-Chem v2.0 and v1.0 successfully captures the abrupt ozone gradient transition in the upper troposphere-lower stratosphere (UTLS) region at tropics and mid latitudes region, demonstrating an excellent agreement with observed concentrations. BCC-GEOS-Chem v2.0 achieves superior representation of vertical ozone structure from the troposphere through the low stratosphere across tropics, Japan, Europe and Canada, showing substantially reduced mean model biases (-21.2−1.6 ppbv at 400-200 hPa) compared with BCC-GEOS-Chem v1.0 and BCC-AGCM-Chem. However, both BCC-GEOS-Chem v2.0 and v1.0 tend to underestimate UTLS ozone levels across Antarctic, reflecting deficiencies in simulating stratospheric ozone at high latitudes, while BCC-AGCM-Chem shows better agreement with observations as its chemical species are set/relaxed to prescribed



monthly climatological values as it does not directly simulate stratospheric chemistry. We also find that BCC-GEOS-Chem
v2.0, which incorporates the comprehensive stratospheric chemistry of the UCX mechanism, better simulates Antarctic
stratospheric ozone depletion in austral spring compared with BCC-GEOS-Chem v1.0, which employs only a simplified
linearized ozone parameterization (LINOZ) (Figure S4).

At the surface, however, all three models tend to overestimate the surface ozone, which has been a long-standing issue
among global chemical models (Gao et al., 2025). Figure 4 shows the scatter plots of the simulated and observed surface
ozone concentrations over Europe, Asia, and the US over 2012-2014.  We only use remote or rural sites here as a global
model at a coarse resolution of ~2.8° is difficult to resolve pollutant levels at urban sites. BCC-GEOS-Chem v2.0 tends to
overestimate surface ozone levels, especially in Asia with high biases of 10-20 ppbv, these high biases are not prominent in
BCC-GEOS-Chem v1.0 and other previous studies using earlier GEOS-Chem model version (e.g. version 11). This
discrepancy may partly attribute to the integration of updated aromatic chemistry in GEOS-Chem models from version
13.0.0 onwards (Lu et al., 2024), which has been shown to elevate surface ozone concentrations by at least 5 ppbv in eastern
China (Bates et al., 2021). Even though the comparisons are limited to rural or remote sites, the low resolution of these
models (~200km) cannot resolve the heterogeneity of ozone precursors, thus lead to artificial mixing and biased ozone
production efficiency (Wile and Prather, 2006; Yu et al., 2016; Young et al, 2018). In addition, these models are difficult to
represent local meteorological conditions particularly over complex terrain at such coarse resolution, further limit their
ability to capture site-level ozone concentrations. We will demonstrate in Section 6 that increasing the spatial resolution can
significantly reduce the bias in simulated surface pollutant concentrations at site level. The larger ozone positive bias in
BCC-GEOS-Chem v2.0 relative to BCC-GEOS-Chem v1.0 may also attribute to lower dry deposition velocity in the model,
as will be discussed in Section 4.3.






**Figure 3. Comparisons of model simulated annual mean ozone vertical profiles to ozonesonde observations averaged over 2012-2014. Black solid line represents WOUDC observations, the others represent the model from BCC-AGCM-Chem, BCC-GEOS-Chem v1.0 and BCC-GEOS-Chem v2.0, respectively. Black horizontal bars are the standard deviations of WOUDC observations. Each panel represents the average of all sites in this region. Numbers of selected sites for each region in 2012–2014 are given.**




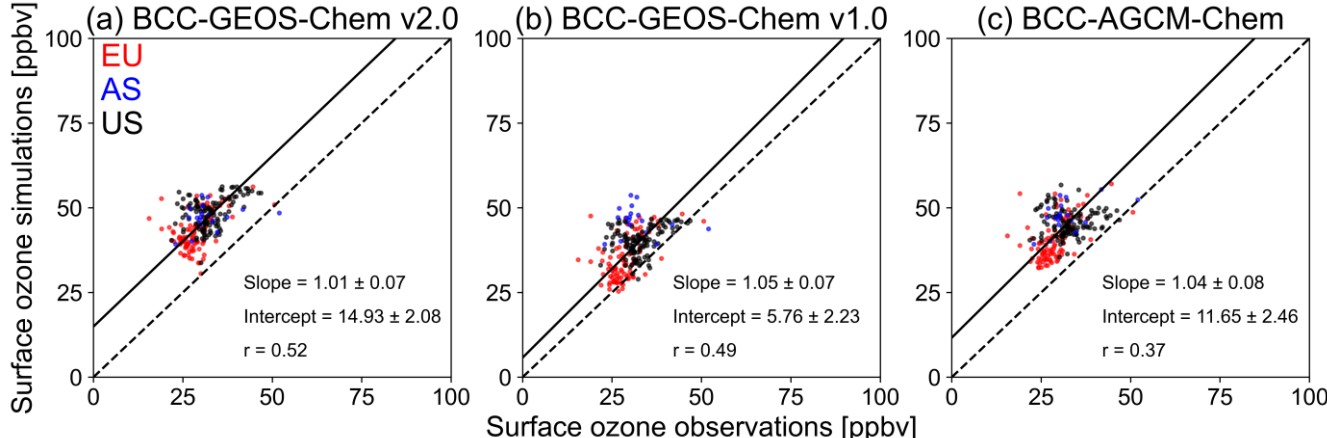

**Figure 4. Scatter plots of observed and simulated surface ozone for (a) BCC-GEOS-Chem v2.0, (b)BCC-GEOS-Chem v1.0, and (c) BCC-AGCM-Chem over 2012-2014. The solid lines indicate the regression lines. The dotted lines indicate the 1: 1 line.**
**Observations are averaged to the same model grid.**

### 4.2 CH$_2$O, NO$_2$, CO, and aerosols

Comparison with satellite observations in Figure 5 demonstrate that BCC-GEOS-Chem v2.0 has significant improvements in simulating tropospheric CH$_2$O, NO$_2$, and CO. BCC-GEOS-Chem v2.0 shows a negative mean bias to OMI tropospheric
CH$_2$O column of $-1.2\pm1.5\times10^{15}$ molecule cm$^{-2}$ averaged over 2012-2014, with positive bias at the tropics. In comparison, both BCC-GEOS-Chem v1.0 and BCC-AGCM-Chem show substantially larger positive bias over the Amazon, central Africa, tropical Asia, and the southeastern United States. This may reflect a larger emission of BVOCs in these two models (isoprene emissions of 410 Tg yr$^{-1}$) compared with BCC-GEOS-Chem v2.0 (389 Tg yr$^{-1}$). The negative bias in tropospheric CH$_2$O column in BCC-GEOS-Chem v2.0 across midlatitude regions of both hemispheres, which is also present in the other
two models examined, could partially originate from systematic uncertainties inherent in satellite retrieval methodologies (Zhu et al., 2016).

For NO$_2$, BCC-GEOS-Chem v2.0 show no significant bias compared with the OMI tropospheric NO$_2$ column when averaged over the globe ($0.0\pm1.3\times10^{15}$ molecule cm$^{-2}$), but this reflects the compensation of negative bias over emission
hotspots such as East Asia, India, West Europe and central Africa, and positive bias over other regions. In comparison, simulated tropospheric NO$_2$ column in BCC-GEOS-Chem v1.0 and BCC-AGCM-Chem show substantially high positive biases of $4$-$7\times10^{14}$ molecule cm$^{-2}$ (55.4-107.3%) over continental lands, especially over East Asia and India. As total NO$_x$ emissions are comparable among the three models, the improved performance on simulated tropospheric NO$_2$ column is more likely attributed to the difference in tropospheric chemistry and deposition.




For CO, we evaluate simulated CO concentrations at 700 hPa where the MOPITT satellite has generally high sensitivity (Emmons et al., 2004; Pfister et al., 2005). BCC-GEOS-Chem v2.0 has also substantially reduced the positive bias relative to observed values, averaged 4.9±11.9 ppbv (5.5%) over the globe, compared with the excessive positive CO bias of BCC-GEOS-Chem v1.0 (24.8±12.9 ppbv) and BCC-AGCM-Chem (50.8±14.1 ppbv) with even larger bias over Asia, central

Africa, and the East Pacific Ocean. The high bias in BCC-AGCM-Chem is probably caused by the excessive emissions from biomass burning and anthropogenic sources (1140 Tg yr$^{-1}$ compared with 925 Tg yr$^{-1}$, Table S1) and a low global column-weighted mean OH concentration ($0.5\times10^6$ molecule cm$^{-3}$).

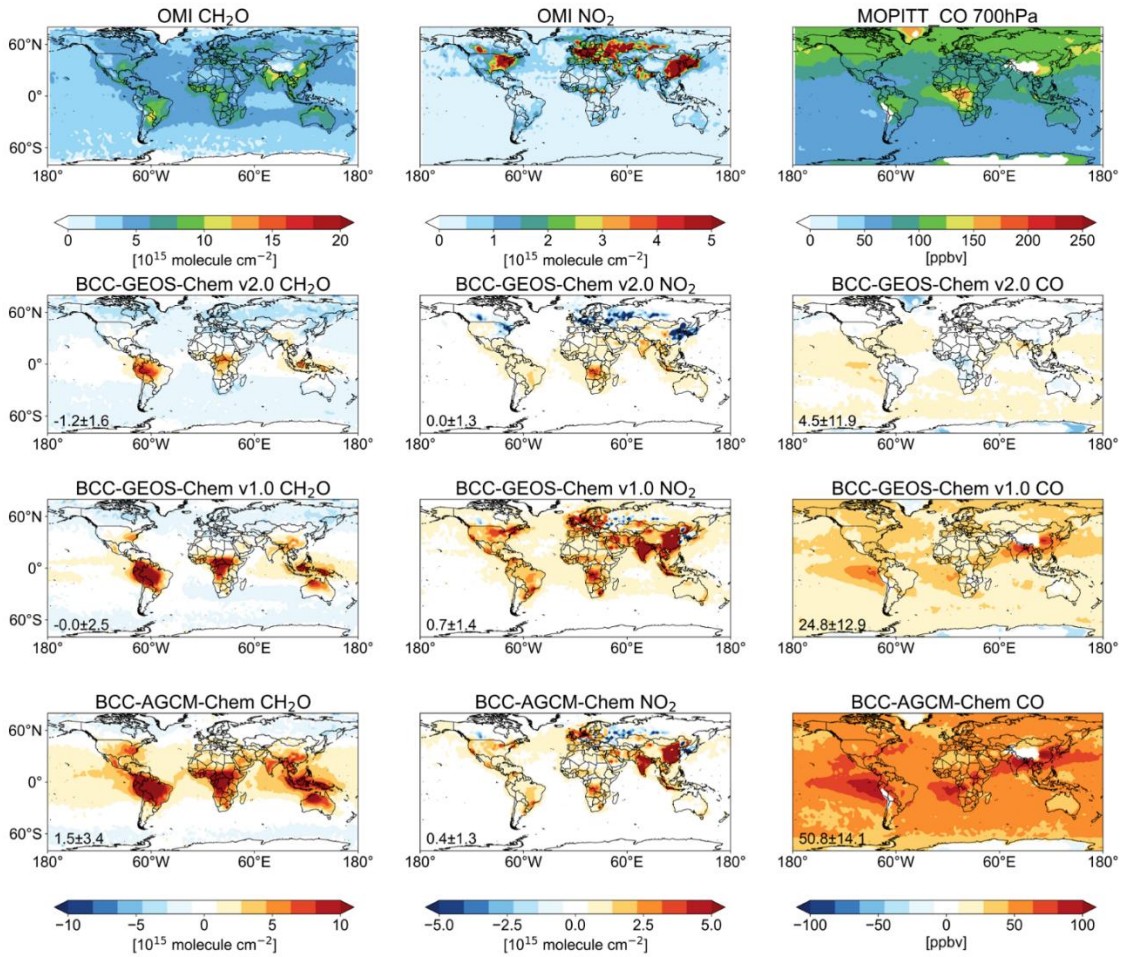

**Figure 5. Spatial distributions of global tropospheric NO₂ column averaged over 2012–2014 from (a) OMI satellite observation, (b) differences between BCC-GEOS-Chem v2.0 and OMI, (c) differences between BCC-GEOS-Chem v1.0 and OMI, and (d) differences between BCC-AGCM-Chem and OMI.**



Figure 6 evaluates the simulated surface fine particulate matter ($PM_{2.5}$) concentrations with the satellite-derived data as
introduced in Section 3.2(Shen et al., 2024). The satellite-derived data show high $PM_{2.5}$ concentration over East Asia and
India due to intensive anthropogenic emissions, and over northern and central Africa due to mineral dust and biomass
burning emissions. BCC-GEOS-Chem v2.0 reproduces the overall spatial distributions and magnitude of observed $PM_{2.5}$
concentrations, though it tends to marginally underestimate concentrations in the hotspot regions (e.g. Amazon, central
Africa, and India), which is most likely due to the coarse model resolution, and the uncertainties of biomass burning
(Reddington et al., 2017). BCC-GEOS-Chem v2.0 demonstrates superior performance in simulating $PM_{2.5}$ concentration
compared with BCC-GEOS-Chem v1.0, which exhibits pronounced overestimation in East Asia, India, the Arabian
Peninsula, and Northern Africa, and BCC-AGCM-Chem, which significantly underestimates $PM_{2.5}$ concentrations in major
emission hotspot regions. This improvement is likely due to its better performance in simulating the gas-phase precursor as
shown above (Figure 5), as well as its more comprehensive representation of aerosol chemical mechanism. The large
negative bias in BCC-AGCM-Chem may be because of the absence of nitrate and SOA formation in the model (Wu et al.,
2020).

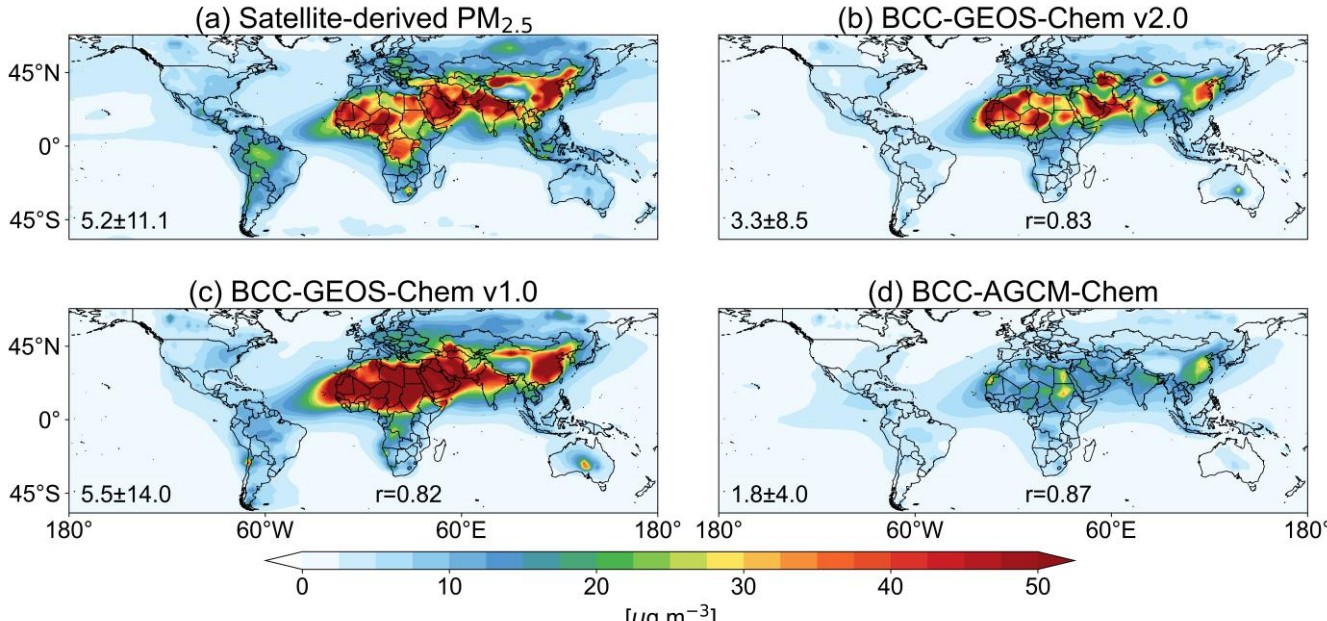

**Figure 6. Spatial distributions of annual mean Surface fine particulate matter ($PM_{2.5}$) from (a) Satellite-derived $PM_{2.5}$, (b) BCC-**
**GEOS-Chem v2.0, (c) BCC-GEOS-Chem v1.0, and (d) BCC-AGCM-Chem. Values are averaged over 2012-2014.**



### 4.3 Diagnostics of tropospheric ozone budgets and OH concentrations

We then diagnose the global tropospheric ozone budget and tropospheric oxidation capacity (using global air-mass-weighted tropospheric OH and methane lifetime against tropospheric OH loss as key metrics) in BCC-GEOS-Chem v2.0, and
compared these results with those from BCC-GEOS-Chem v1.0 and other studies in the literature (Table 4). The results from BCC-AGCM-Chem are not presented here, as its diagnostics for $O_x$ production and loss is not available.

The global tropospheric ozone burden simulated by BCC-GEOS-Chem v2.0 is 355 Tg averaged over 2012-2014, about 5.7% higher than the result from BCC-GEOS-Chem v1.0 and falls within the range (250-410 Tg) estimated by ~50 models
assessments from Young et al. (2018). The global tropospheric mean OH concentration in BCC-GEOS-Chem v2.0 is smaller than that from BCC-GEOS-Chem v1.0 ($0.97\times10^6$ vs. $1.16\times10^6$ molecule cm$^{-3}$) and falls within the range ($0.74$–$1.33\times10^6$ molecule cm$^{-3}$) estimated from multiple chemical models (Naik et al., 2013). Table 3 further compares the simulated air-mass-weighted tropospheric mean OH with the observation-based results from Zhao et al. (2023) across different latitudinal bands. The observation-based OH concentrations are estimated by adjusting simulated OH precursors and meteorological
conditions to the observations in chemical box models. We find that the overall latitudinal distributions of OH in BCC-GEOS-Chem v2.0 align with observation-based results with biases below $2\times10^5$ molecule cm$^{-3}$, whereas BCC-GEOS-Chem v1.0 has a significant high bias, especially in the tropics ($3.9$-$4.8\times10^5$ molecule cm$^{-3}$). Discrepancies in meteorological condition and concentrations of methane, ozone, $NO_x$, and CO can all contribute to the OH bias in climate–chemistry models (Nicely et al., 2020). The derived methane chemical lifetime against OH loss is 9.28 years in BCC-GEOS-Chem v2.0, which
falls in the middle range reported from multi-model assessments (7.1-14.0 years) (Naik et al., 2013).

We further examine the diagnostics of the chemical production and loss of the odd oxygen ($O_x$) family, which includes $O_3$, $NO_2$, $NO_y$, several organic nitrates, and halogen species (a total of 67 species in GEOS-Chem v14.0.1) to account for rapid cycling between $O_x$ components. Ozone accounts for over 95% of total $O_x$ according to Hu et al. (2017). The global annual
ozone chemical production and loss are 6356 and 6272 Tg, respectively. These estimates of $O_x$ chemical production and loss are higher than results in BCC-GEOS-Chem v1.0 but still fall in the range from ~30 models assessments presented by Young et al. (2018). However, we find that the difference between the estimated $O_x$ chemical production and loss (~100 Tg) is much smaller than theses multi-model estimate (~500 Tg) (Table 4). GEOS-Chem "Classic" version 14.0.0 driven by reanalysis meteorology fields also yields a relatively small difference between the $O_x$ chemical production and loss of 377 Tg
(https://ftp.as.harvard.edu/gcgrid/geos-chem/1yr_benchmarks/). The production to loss difference between the BCC-GEOS-Chem v2.0 and multi-model estimates in Young et al. (2018) may be partly attributed to the discrepancies in the metric ($O_x$ vs ozone), but is more likely due to the difference in chemical scheme.



The global annual mean ozone dry deposition flux simulated by BCC-GEOS-Chem v2.0 is 1061 Tg averaged over 2012-
2014, which shows agreement in the results of BCC-GEOS-Chem v1.0 and falls within the range (700-1500Tg) estimated by
Young et al. (2018). Figure 7 illustrates the annual mean distribution of global ozone dry deposition velocity over 2012–
2014, as simulated by BCC-GEOS-Chem v2.0, BCC-GEOS-Chem v1.0, and BCC-AGCM-Chem. The overall spatial
distribution of ozone dry deposition velocity in BCC-GEOS-Chem v2.0 is similar to that in BCC-GEOS-Chem v1.0, but the
values are slightly slower. This is attributed to two reasons. First, as introduced in Section 2.3.3, BCC-GEOS-Chem v2.0
utilizes offline Olson map land types and LAI data to replace that from the BCC-AVIM. In contrast, BCC-GEOS-Chem v1.0
uses land types and leaf area indices information diagnosed from the BCC-AVIM. Second, the BCC-GEOS-Chem v2.0 has
included the new deposition scheme over oceans (Pound et al., 2020). The new scheme has shown to reduce the area-
weighted mean ozone deposition over ocean by 50%, and the global total deposition flux by 17% (Pound et al., 2020).
However, the total ozone dry deposition flux in BCC-GEOS-Chem v2.0 is 20% higher than that in BCC-GEOS-Chem v1.0,
due to the higher surface ozone concentration. In comparison, the BCC-AGCM-Chem applies the archived ozone dry
deposition velocity from MOZART simulation (Wu et al., 2020), which is likely too small compared with observations.

We estimate the annual STE of ozone as 977 Tg in BCC-GEOS-Chem v2.0, calculated as the residual of the mass balance
among tropospheric chemical production, chemical loss, and deposition. This is higher than the high end of the range from
multi-model estimates (180-920 Tg) in Young et al. (2018). The estimate of BCC-GEOS-Chem v1.0 (336 Tg) was
considered too small and caused the ozone underestimates in the upper troposphere (Figure 3), primarily due to the lack of
representation in stratospheric chemistry (Lu et al., 2020). However, the inclusion of UCX in BCC-GEOS-Chem v2.0 seems
to over-correct this issue. Figure 3 shows that in the UTLS regions, BCC-GEOS-Chem v2.0 agrees well with the ozonesonde
observations in the tropics, East Asia, Canada, and Europe, while shows high ozone bias in the US and low bias Antarctica.
However, we do find that BCC-GEOS-Chem v2.0 shows high ozone bias in the middle stratosphere compared with the
archived ozone concentration used as forcing data in CMIP6, although it is able to capture the main features in stratospheric
chemistry such as the Antarctica ozone depletion as introduced in Section 4.1. The high stratospheric ozone bias may be at
least partly attributed to the relative low model top in BCC-GEOS-Chem (3 hPa), which limits the inclusion of upper
boundary conditions for $NO_x$ and halogens from mesosphere. Consequently, the stratospheric ozone is overestimated due to
the underestimated levels of halogens and $NO_x$ that deplete ozone. The STE diagnosed from the mass balance method may
also be biased as it is not a direct diagnostic of the dynamic flux between stratosphere and troposphere. More work is
required to address this issue.





**Table 3.** The simulated and observation-based OH (in $10^5$ molecule cm$^{-3}$) averaged over latitudinal bands.

|  | 90-30° S | 30-0° S | 0-30° N | 30-90° N |
|---|---|---|---|---|
| Observation-based [a] | 5.3 | 12.2 | 14.5 | 7.2 |
| BCC-GEOS-Chem v2.0 | 4.6 | 12.1 | 12.7 | 6.0 |
| BCC-GEOS-Chem v1.0 | 7.0 | 17.0 | 18.4 | 9.0 |

[a] From Zhao et al. (2023) simulations in CESM constrained by observations.


**Table 4.** Global Budget Diagnostics of Tropospheric Ozone in BCC-GEOS-Chem v2.0

| Diagnostic term related to tropospheric ozone | BCC-GEOS-Chem v2.0 (this study) | BCC-GEOS-Chem v1.0[a] (Lu et al., 2020) | References |
|---|---|---|---|
| Ozone burden (Tg) | 355 | 336 | Mean: 340; range: 250–410[b] |
| $O_x$ chemical production (Tg yr$^{-1}$) [c] | 6356 | 5486 | Mean: 4900; range: 3800–6900[d] |
| $O_x$ chemical loss (Tg yr$^{-1}$) [c] | 6272 | 4983 | Mean: 4600; range: 3300–6600[d] |
| Dry deposition (Tg yr$^{-1}$) | 1061 | 873 | Mean: 1000; range: 700–1500[d] |
| Global OH ($10^6$ molecule cm$^{-3}$) [e] | 0.97 | 1.16 | Mean: 1.11; range: 0.74–1.33[f] |
| Methane chemical lifetime (yr) | 9.28 | 8.27 | Mean: 9.7; range: 7.1-14.0[f] |
| STE (Tg yr$^{-1}$) [g] | 977 | 370 | Mean: 500; range: 180–920[d] |

[a] From Lu et al. (2020) results in BCC-GEOS-Chem v1.0, the version of GEOS-Chem was v11-02. [b] Based on Figure 3 in Young et al. (2018), the ensemble burden represents ~50 models for year 2000 conditions. [c] Budget pertains to the odd oxygen ($O_x$) family, which includes $O_3$, $NO_2$, $NO_y$, several organic nitrates, and halogen species (a total of 67 species in GEOS-Chem v14.0.1). Ozone constitutes more than 95% of the total $O_x$. [d] From Figure 3 in Young et al. (2018), the ensemble fluxes represent ~30 models for year 2000 conditions. [e] Global annual mean air mass weighted OH concentration in the troposphere.
[f] From Table 1 in Naik et al. (2013), the budget is based on an ensemble of 16 models for year 2000 conditions. [g] Estimated from the residual of mass balance among tropospheric chemical production, chemical loss, and deposition.


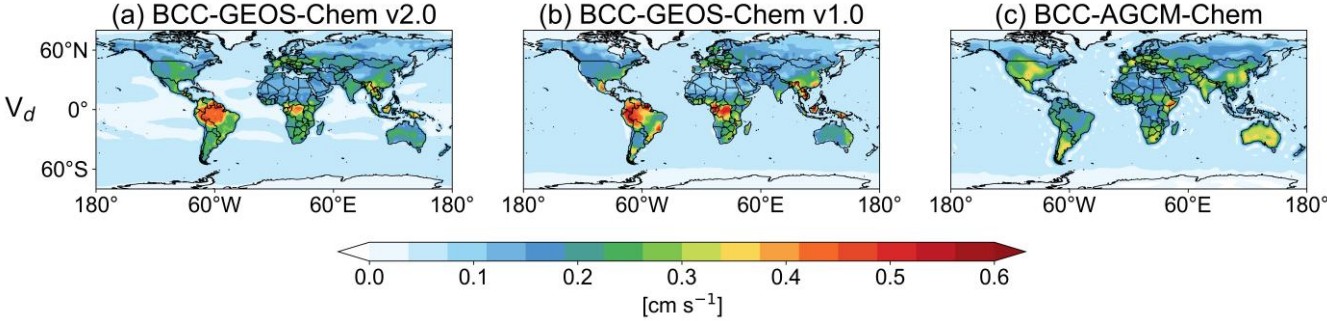

**Figure 7. Annual mean distribution of global ozone dry deposition velocity for 2012–2014 from (a) BCC-GEOS-Chem v2.0, (b) BCC-GEOS-Chem v1.0, and (c) BCC-AGCM-Chem.**




## 5 Influence of aerosols on radiation and clouds in BCC-GEOS-Chem v2.0

In this section, we evaluate the capability of the BCC-GEOS-Chem v2.0 model to simulate key metrics describing climate conditions, and quantify the contributions of atmospheric aerosols to these metrics as a means to assess the implementation

of aerosol feedback in the model. These metrics to be examined include surface temperature (TS), precipitation, radiative and cloud properties such as downward shortwave (SWDOWN) radiation, outgoing longwave radiation (OLR), shortwave and longwave cloud radiative forcing (SWCRF and LWCRF), cloud fraction, cloud droplet number concentration, and cloud liquid water. Model performance is evaluated through comparisons with satellite observations and MERRA-2 reanalysis data (Table S3). Results are summarized in Figures 8–9 and Table 5.


Global annual mean surface temperature in BCC-GEOS-Chem v2.0 is 288.5 K, slightly higher than 287.1 K in MERRA-2 reanalysis data averaged over 2012-2014. BCC-GEOS-Chem v2.0 also demonstrates robust performance in simulating radiative fluxes and cloud properties, with consistent spatial distributions and no significant bias compared with observations or reanalysis data. The simulated global annual mean SWDOWN and OLR shows general agreement with CERES data

(153.9 vs. 164 W m$^{-2}$ for SWDOWN and 217.3 vs. 224.9 W m$^{-2}$ for OLR, respectively), with high spatial correlation coefficients ($r$) exceeding of 0.96 between the simulation and CERES data. The model also reproduces the MERRA-2 global annual mean CF (0.65 vs. 0.65) and its spatial distribution ($r$=0.66). Simulated total precipitation also aligns with GPCP data (2.33 vs. 2.24 mm day$^{-1}$) with a spatial correlation coefficient of 0.83. Slightly larger bias is found for cloud radiative forcing. The simulated SWCRF underestimates CERES observations by 9.8 W m$^{-2}$ (-39.8 vs. -49.2 W m$^{-2}$), while LWCRF is

overestimated by 3.8 W m$^{-2}$ (26.4 vs. 22.6 W m$^{-2}$). The cloud liquid water mass fraction shows excellent agreement with MERRA-2 reanalysis data (5.92 × 10$^{-6}$ vs. 5.95 × 10$^{-6}$ kg kg$^{-1}$).

The BCC-GEOS-Chem v2.0 model is able to reproduce the aerosol's impact on radiative budget, e.g., decreasing SWDOWN, OLR, and LWCRF, and increasing SWCRF (Table 5), as reported in literatures (e.g. Pendharkar et al., 2023).

Sensitivity simulations further quantify the impact of ARIs and ACIs on the radiation (Figure 8 c and d). ARIs and ACIs lead to changes in SWDOWN of -3.3 and -14.1 W m$^{-2}$, respectively. The spatial heterogeneity in ARI-induced changes in SWDOWN is mainly attributed to the absorbing characteristics of aerosols and their semi-direct effects on cloud droplet evaporation. The impact of ACIs on SWDOWN is mainly driven by clouds activated by aerosols, leading to a greater scattering of shortwave radiation. These results are consistent with regional model studies across Asia and Africa (Feng et al.,

2021; Feng et al., 2025) and global climate model studies (Wang et al., 2021; Zhao et al., 2022). For OLR, we find that ARIs decrease the global mean value by 0.6 W m$^{-2}$ but with a mild increase in tropical regions where absorbing aerosols are higher, while ACIs substantially decrease by OLR by 9.7 W m$^{-2}$, largely attributable to its impact on high clouds. In addition, the model estimates that ACIs amplify SWCRF by 14.6 W m$^{-2}$ while decrease LWCRF by 1.3 W m$^{-2}$, showing higher impacts



than ARIs. These tendencies are consistent with reported aerosols' feedback on cloud radiative forcing (Pendharkar et al.,
635    2023).

Figure 9c and d further quantify the impact of aerosols on cloud properties and total precipitation through ACIs and ARIs
from BCC-GEOS-Chem v2.0. For global average, ACIs promote cloud development but suppress precipitation, as evident in
an increase of 0.2% in cloud fraction and 21.7% in cloud liquid water but a decrease of 16.3% in total precipitation in the
BASE simulation compared with the noACIs simulation (Table 5). The increases in cloud fraction and cloud liquid water are
notable in the middle and high latitudes, while the influences on precipitation are mostly concentrated in the tropics,
consistent with other modelling studies (e.g. Zelinka et al., 2014; Pendharkar et al., 2003). These results reflect the Twomey
effect (Twomey 1977), i.e., aerosols are activated as cloud condensation nuclei (CCN), more but smaller cloud droplets are
activated. With the reduction of the effective radius of cloud droplets, the collision-coalescence processes are hindered, and
the precipitation of cloud water is delayed (Rosenfeld et al., 2008). Our simulations thus suggest reasonable feedback of
aerosol indirect effect in BCC-GEOS-Chem v2.0. In comparison, the impact of ARIs on cloud properties and precipitation
are rather small.

**Table 5.** Comparison of the radiative and cloud properties between observations and simulations.

| Variable | Observation/Reanalysis | Base | noARIs | noACIs |
|---|---|---|---|---|
| SWDOWN | 164.0 | 153.9 | 157.2 | 168.0 |
| TS | 287.1 | 288.5 | 289.1 | 293.5 |
| OLR | 224.9 | 217.3 | 217.9 | 227.0 |
| SWCRF | -39.8 | -49.2 | -51.1 | -34.6 |
| LWCRF | 22.6 | 26.4 | 26.5 | 27.7 |
| CF | 0.65 | 0.65 | 0.65 | 0.65 |
| CDNC | / | 69 | 69 | 5(fixed) |
| CLW ($10^{-6}$) | 5.92 | 5.95 | 5.99 | 4.66 |
| PREP | 2.24 | 2.33 | 2.36 | 2.71 |




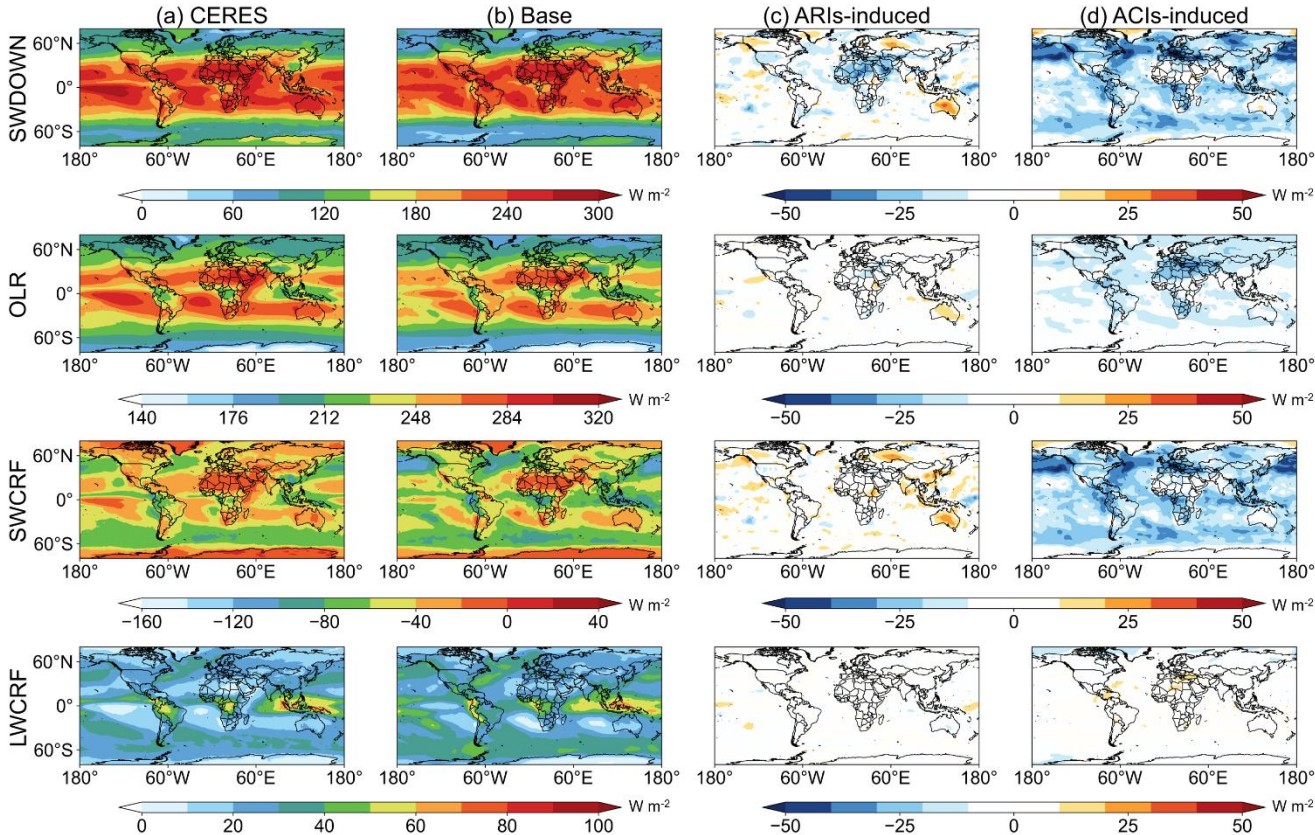

**Figure 8.** Annual mean radiative properties from (a) CERES satellite observations, (b) Base simulation, (c) ARIs-induced, and (d) ACIs-induced. Variables evaluated include the downward shortwave (SWDOWN) radiation, shortwave cloud radiative forcing (SWCRF), longwave cloud radiative forcing (LWCRF), and outgoing longwave radiation (OLR).





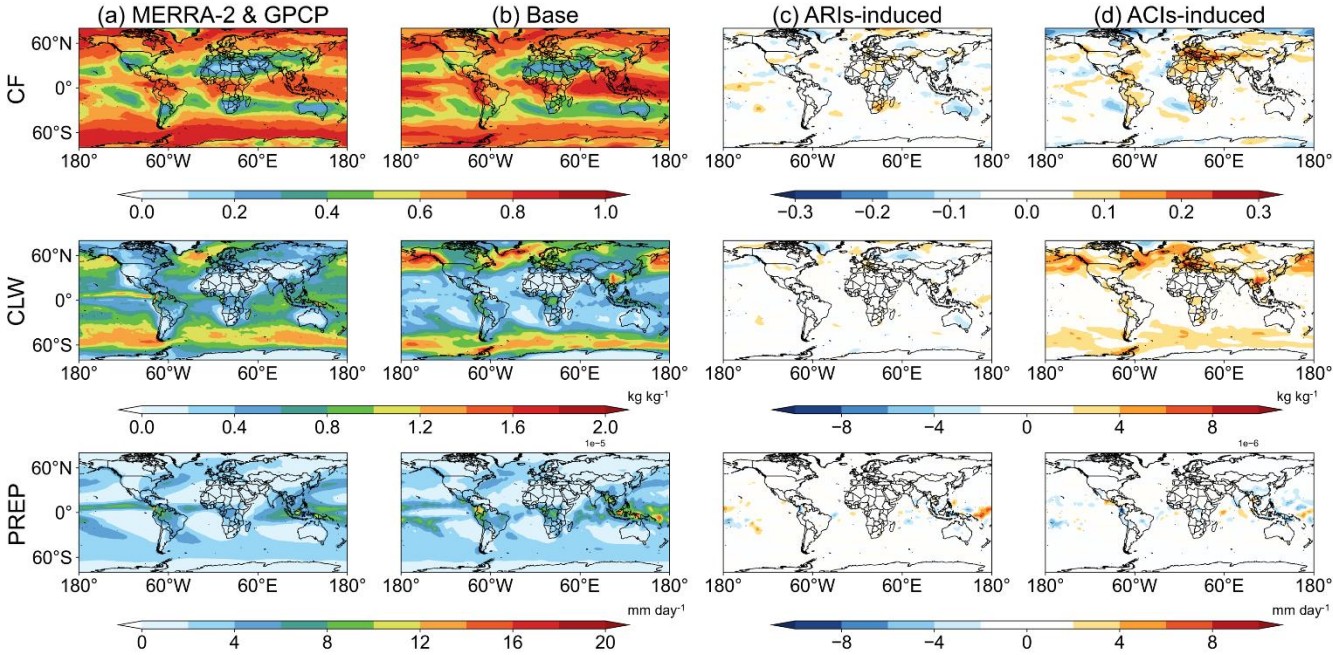

**Figure 9. Annual mean cloud properties from (a) MERRA-2 reanalysis data or GPCP satellite data, (b) Base simulation, (c) ARIs-induced, and (d) ACIs-induced. Variables evaluated include the cloud fraction (CF), mass fraction of cloud liquid water (CLW), and total precipitation (PREP).**

## 6 High-resolution simulation capability in BCC-GEOS-Chem v2.0: comparison of one-month simulation between T42L26 and T159L72

Our analysis above has demonstrated the capability of BCC-GEOS-Chem v2.0 in simulating atmospheric chemistry and climate at T42L26 resolution. In this section, we evaluate the high-resolution (T159L72) configuration of BCC-GEOS-Chem v2.0, with approximately 0.75° latitude × 0.75° longitude for horizontal resolution and with 72 vertical hybrid layers extending from the surface to 0.01 hPa. This further utilizes the grid-independent coupling capability of BCC-GEOS-Chem, enabling seamless integration of the GEOS-Chem chemical module with ESMs across customizable grid configurations and spatial resolutions. As such, the chemical module in BCC-GEOS-Chem v2.0 for the two configurations are exactly the same. The T159L72 simulation was constrained to a one-month testing period (January 2014) due to the substantial computational cost escalation, which demands approximately 104 times more computational resources than the T42L26 simulation.

Figure 10 compares the spatial distributions of global atmospheric CO concentrations at 700 hPa and surface $NO_2$, ozone, and $PM_{2.5}$ concentrations in China from BCC-GEOS-Chem v2.0 simulation at T159L72 versus T42L26 resolution. Comparison of CO concentrations at 700 hPa aims to demonstrate the differences in transport of trace gases in the free



troposphere between the simulations at the two resolutions. As expected, the T159L72 configuration demonstrates markedly
enhanced capability in resolving fine-scale CO plume transport dynamics, particularly pronounced over African and Indonesian source regions, whereas the corresponding coarse-resolution T42L26 simulation fails to reproduce these persistent plumes due to rapid dissipation by numerical diffusion (Eastham et al., 2017; Zhuang et al., 2018) (Figure 10a and b). These results highlight the advantage of high-resolution simulation in mitigating numerical diffusion in Eulerian models.

Comparison of surface $NO_2$, ozone, and $PM_{2.5}$ concentrations aims to reveal the distinct resolution-dependent characteristics in pollution patterns. We thus focus on China where air pollutions were severe in January 2014. For surface $PM_{2.5}$ and $NO_2$ concentrations, the T159L72 configuration successfully resolves concentrated pollution hotspots over eastern China (Figure 10d, f, and h). In contrast, the coarser T42L26 model significantly attenuates these peak values through excessive spatial smoothing (Fig 10c, e, and g). This disparity arises from the inability of model at coarse resolution to resolve fine-scale
emission features from the inventory data, coupled with artificial mixing artifacts inherent to large grid-cell discretization. For surface ozone simulations, the T159L72 configuration produces higher ozone concentrations across most regions compared with the coarser T42L26 simulation, while showing lower ozone levels over the North China Plain. This is attributable to the unique photochemical regime in wintertime North China, where substantial $NO_x$ emissions drive ozone formation into a VOC-limited regime. The intensified $NO_x$ titration effect under such chemical conditions effectively
suppresses ozone accumulation, a process that the high-resolution model better represents through refined precursor gradient resolution. Comparison of the simulated surface $PM_{2.5}$, $NO_2$, and ozone concentrations with surface monitoring network of air quality in China (Figure 10) supports that the T159L72 simulation shows higher spatial correlation coefficients (r=0.40-0.55) with the observed values and substantially reduce the mean biases (MB) cross all sites (expect for ozone) compared with the T42L26 simulation. In summary, the high-resolution implementation of BCC-GEOS-Chem v2.0 highlights the
advantages and enhanced flexibility of our BCC-GC-HEMCO interface, and shows attractive potential for future applications in air quality simulation.







**Figure 10. Spatial distributions of BCC-GEOS-Chem v2.0 simulated global 700hPa CO, and surface NO₂, surface O₃, and surface PM₂.₅ in China at the resolution of T42L26 and T159L72 in January 2014. Mean bias (MB) and spatial correlation coefficients between the simulated and observed values are shown inset.**






## 7 Summary and future plans

We have developed the BCC-GEOS-Chem v2.0, an online two-way chemistry-climate coupled model that incorporates feedback from chemically reactive greenhouse gases and aerosols. BCC-GEOS-Chem v2.0 has evolved into a fully coupled

Earth system model that integrates the modules including the atmospheric component BCC-AGCM, the land component BCC-AVIM, the oceanic component MOM, and sea ice component SIS. The standard chemical mechanism in the BCC-GEOS-Chem v2.0 features a comprehensive $O_x$-$NO_x$-VOC-halogen-aerosol chemistry in unified tropospheric–stratospheric chemistry extension (UCX) scheme. The BCC-GC-HEMCO interface serves as a nexus linking the atmospheric module, an independent HEMCO emission module, and the GEOS-Chem chemical module.


We use satellite, ozonesonde, and surface observations to evaluate the performance of the BCC-GEOS-Chem v2.0 simulation at T42L26 ($2.8° \times 2.8°$ and 26 vertical layers with a top at 2.914 hPa) resolution in representing atmospheric chemistry, and compare with the model outputs from the BCC-GEOS-Chem v1.0 and BCC-AGCM-Chem for the same simulated time period (2012–2014). Results demonstrates that BCC-GEOS-Chem v2.0 captures well the main spatial and

seasonal distributions of tropospheric ozone observed by multiple instruments, and also shows improved performance in simulating major gaseous pollutants and surface $PM_{2.5}$ compared with the BCC-GEOS-Chem v1.0 and BCC-AGCM-Chem. The diagnostics of tropospheric ozone budgets and OH concentrations generally agree with the observation-constrained values and multi-model assessment. However, the model derives a relative low value of the difference between tropospheric ozone chemical production and loss and shows a high STE flux that requires further investigation.


BCC-GEOS-Chem v2.0is now able to consider aerosol-climate interactions through radiation and cloud feedback, based on the diagnosed mass mixing ratios of bulk aerosols. The base simulation shows that BCC-GEOS-Chem v2.0 with full aerosol–cloud–radiation interactions reproduce the global spatial distributions of radiation and cloud properties from satellite observations and reanalysis dataset. Sensitivity experiments indicate that the model is able to simulate the impact of aerosols

on radiative budget, e.g., decreasing SWDOWN, OLR, and LWCRF, and increasing SWCRF.   Further analyses by separating the role of ARIs and ACIs indicate that ACIs promote cloud development but suppress precipitation for global average, while the contributions from ARIs are rather small.

BCC-GEOS-Chem v2.0 can now perform at a high resolution at T159L72 ($0.75° \times 0.75°$ and 72 vertical layers with a top at

0.01 hPa), taking the advantages of the BCC-GC-HEMCO interface that enables seamless integration of the GEOS-Chem chemical module with ESMs across customizable grid configurations. Compared with the T42L26 simulation, the high-resolution simulation much better resolves the fine-scale plume transport dynamics, and also significantly improve the ability to capture the pollution hotspot of $NO_2$ and $PM_{2.5}$ as well as the low ozone concentration in high-$NO_x$ environment in wintertime China. The shows attractive potential for future applications in air quality simulation.




In summary, the development of the BCC-GEOS-Chem v2.0 model provides a powerful tool to study climate-chemistry interactions and for future projection of global atmospheric chemistry. Its high-resolution configuration can also be applied to air quality forecast. Nevertheless, there are still limitations in this model that should be addressed in future development. First, although the UCX interactive stratospheric chemistry scheme has been implemented, biases remain in the simulation of

stratospheric ozone and will require further improvement. In addition, the implementation of size-resolved aerosol schemes will be a priority moving forward, as they will enable a more accurate representation of aerosol feedbacks. Furthermore, the enormous amount of time and computational resources for T159L72 high-resolution is still a major obstacle. We are planning to improve this high-resolution version of BCC-GEOS-Chem v2.0 with enhanced computational efficiency in our next-phase implementation, enabling its broader application in research of global climate change and atmospheric chemistry.


*Code availability.* The GEOS-Chem model is maintained at the Harvard Atmospheric Chemistry Modelling group, which can be obtained at https://doi.org/10.5281/zenodo.7271960 (last access: 1 April 2025). The BCC-GEOS-Chem v2.0 coupler can be accessed at a DOI repository https://zenodo.org/records/16734855, and model outputs for 2012–2014 are available at

https://zenodo.org/records/16734757. All source code and data can also be accessed by contacting the corresponding authors Lin Zhang (zhanglg@pku.edu.cn).

*Author contributions.* LZ designed and led the project. RZS and XL developed the model source code with substantial contributions from HPL, WTW, JZ, and FZ. RZS performed model simulations, analysed data, and prepared the figures with

contributions from XPY, LJY, and, HLW. RZS, XL, and LZ wrote the paper. All authors contributed to the interpretation of results and improvement of the paper.

*Competing interests.* The authors declare that they have no conflict of interest.

*Acknowledgements.* This research was supported by the National Key Research and Development Program of China (2023YFC3706104) and the National Natural Science Foundation of China (grant no. 42275106). This research is also supported by the Guangdong Basic and Applied Basic Research Foundation (2025B1515020034). We thank the GEOS-Chem support team for facilitating the open-source development of GEOS–Chem model.

*Financial support.* This research was supported by the National Key Research and Development Program of China (2023YFC3706104) and the National Natural Science Foundation of China (grant no. 42275106). This research is also supported by the Guangdong Basic and Applied Basic Research Foundation (2025B1515020034).



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

8.
