# Peer review of "Development of the global chemistry-climate coupled model BCC-GEOS-Chem v2.0: improved atmospheric chemistry performance and new capability of chemistry-climate interactions"

_EGUsphere, 2025_

## Referee Comment (RC2)

Review of the manuscript

**Development of the global chemistry-climate coupled model BCC-GEOS-Chem v2.0: improved atmospheric chemistry performance and new capability of chemistry-climate interactions**

submitted to EGUsphere by R. Sun et al.

**General Comments**

This paper describes and evaluates a new Earth System Model, obtained by combining the GEOS-Chem Chemistry-Transport Model (CTM) with the Beijing Climate Centre Earth System Model (BCC-ECSM). The evaluation comprises tropospheric composition, with a focus on the ozone burden and budget, the impacts of Aerosol-Radiation-Interactions and Aerosol-Cloud-Interactions on radiative fluxes and cloud properties, and a high-resolution simulation of pollution above Asia. Since earlier versions of BCC-ESM participated in CMIP5 and in the CMIP6 Aerosol Chemistry Model Intercomparison Project (AerChemMIP; Zhang et al., 2021), we can expect this new version to contribute to future CMIPs as well. It is thus important to document and evaluate this version properly, preferably by publication in Geoscientific Model Development. Yet the submitted manuscript requires major improvements.

The modelling of stratospheric transport and chemistry can be understood only from two unclear sentences in the description of the setup of model experiments (section 3.1), and seems fundamentally flawed:

> "*We note that as BCC-GEOS-Chem only extends to the height of 2.914 hPa, ozone concentrations at the top two layers are set to prescribed monthly climatological values from CMIP6 data package as upper boundary conditions.* [The earlier version] *BCC-AGCM-Chem conducted the same treatment but included additional stratospheric species ($CH_4$, $N_2O$, $NO$, $NO_2$, $HNO_3$, $CO$, and $N_2O_5$), and* [in the current version(?)] *their concentrations from below the top two layers to the tropopause are relaxed at a relaxation time of 10 d towards the climatology.*"

One should first note that this approach precludes any true interactivity between climate change and stratospheric chemistry. This aspect is barely more advanced in BCC-GEOS-Chem v2.0 than in v1.0 and sems actually less developed than in the first version BCC-AGCM-Chem.

As I understand it, the 7 listed species are actually included in the current version, but their distributions are not realistic enough to model stratospheric ozone in a usable manner, and this was partly mitigated by the described relaxation to some (undocumented) climatology. I believe that the model top is too low to allow a proper Brewer-Dobson Circulation, hindering a correct stratospheric composition and explaining the (undocumented) statement that "*BCC-GEOS-Chem v2.0 shows high ozone bias in the middle stratosphere*" (line 575). This would of course explain the very large closure term in the tropospheric ozone budget, i.e. a stratospheric influx residual as large as 977 Tg/year.

In my view the authors have a choice between

- solving this issue by raising the model top, re-running all the experiments, and re-writing all sections related with stratospheric and tropospheric ozone; or…
- accepting this flaw, documenting it in a complete and straightforward manner.

If the model top is raised, I would recommend using the same 72 layers and top at 0.01 hPa as in the high-resolution configuration described in section 6. If not, the high-resolution configuration provides a means to evaluate the impact of the low top in the low-resolution configuration: see SC7 for more ideas on this route. Note that the vertical grid should be documented in the Supplementary material. This high-top vertical grid seems similar to MERRA-2, which has 72 staggered hybrid-pressure levels, i.e. 73 layers. All modern GCMs use staggered vertical grids. Is this also the case for BCC?

In any case, the model evaluation will have to include the vertical distribution of ozone in the stratosphere. Considering the absence of interactive stratospheric chemistry in BCC-GEOS-Chem v1.0 and the "*critical role of stratospheric chemistry in modulating global atmospheric dynamics*" (lines 108-109), the distribution of ozone in the middle atmosphere must be thoroughly documented by quantitative comparisons with observations or at least a climatology of observations. The evaluation of Total Ozone Columns (Fig. S4) is severely lacking as it is purely qualitative, does not include any observational reference and provides no information on the vertical distribution. These comparisons will also provide a basis to report on future progress with the current overestimation of stratospheric ozone influx ("STE", section 4.3).

Please note that the sections "Specific comments" and "Language improvements" end in section 3.1. This reviewer felt that the text requires too many revisions to warrant such a detailed review of the next sections.

**Other major Comments**

**MC1.** The section on code availability is very incomplete as it includes only the "coupler" component between GEOS-Chem and BCC-ESM, and a reference to the generic GEOS-Chem code distribution of the Harvard Atmospheric Chemistry Modelling group. Every module of BCC-GEOS-Chem v2.0 (see Fig. 1) should be covered in this section. If these modules are distributed through different DOI repositories, then their assembly should build "out of the box". For example: can GEOS-Chem be linked directly to the coupler code archived for this paper? If the Geos-Chem modules require some tailored modifications, then this forked code should be distributed by the authors of this paper. What is the code availability for the dynamics, physics and HEMCO modules of BCC-AGCM ?

**MC2.** Climate model evaluations usually rely on mean bias estimates from comparisons between decadal simulations and observational climatologies (for stratospheric composition, see e.g. Froidevaux et al., 2019). When no decadal simulation is available (as seems the case here), the simulations are commonly run in "chemical forecast" mode, with composition running unconstrained but meteorological fields constrained by observations through reanalyses (e.g. MERRA2 or ERA5). Like all CTMs, GEOS-Chem was developed and evaluated in such a configuration. But BCC-GEOS-Chem v2 seemingly lacks this ability, raising the possibility that composition biases are due to unrealistic temperature or circulation patterns. To mitigate this concern, the paper should include, at least in its Supplementary material, all relevant evaluations of temperature and circulation patterns over the evaluated period (2012-2014) and these evaluations should be considered in the discussions of section 4. The introduction should also clarify if BCC-GEOS-Chem v2 can be constrained (or nudged?) by reanalyses, or if there are plans to develop such configurations in the future. This is especially relevant for the Air Quality applications discussed in section 6.

**MC3.** The evaluation and discussion of $CH_2O$, $NO_2$, CO, and aerosols (section 4.2) is too superficial and requires major improvements. The explanations based on biases emission levels are not credible, because these biases are much too small to explain the reported biases in these three species.

**MC4.** The text is often difficult to read and requires English language improvements beyond the suggestions below.

**Specific Comments**

Original text is copied in *italics*.

**SC1.** Lines 50-57: the IPCC reports are not included in the list of references. Please correct this by adding references to the relevant *chapters* in each WG1 Assessment Report. Please cite additional and specific references to support the statement "*most existing CMIP models employ highly simplified representations of atmospheric chemistry*" (lines 54-55).

**SC2.** Line 140: please cite a reference for "*the National Centre for Atmospheric Research (NCAR) flux coupler.*"

**SC3.** Lines 243-244: This sentence is confusing. Please summarize again the aspects of an "*interactive troposphere-stratosphere*" which were missing in BCC-GEOS-Chem v1.0.

**SC4.** Lines 282-283: please list the modelled SLCF explicitly. It looks like this sentence is not about SLCF in general but aerosols specifically.

**SC5.** Lines 299-300, this sentence is very unclear: "*Finally, aerosols across different size bins are assumed to be externally mixing, which are subsequently used in radiative transfer calculation*". Please re-formulate and clarify!

**SC6.** Lines 337-339: "*We note that as BCC-GEOS-Chem only extends to the height of 2.914 hPa, ozone concentrations at the top two layers are set to prescribed monthly climatological values from CMIP6 data package as upper boundary conditions.*" This is a key issue which needs to be expanded (see General Comment). If these Boundary Conditions are kept, where do they come from? What is the second pressure level? Does this approach introduce vertical discontinuities?

**SC7.** Lines 664-665: "*…with 72 vertical hybrid layers extending from the surface to 0.01 hPa.*" See the general comment: if the authors decide to keep the current low top at 2.914 hPa, this higher top in the high-resolution configuration allows at least to check the impact of the low top in the low-resolution configuration. The required evaluation of the vertical distribution of ozone should include the results from this high-resolution simulation in the upper stratosphere. The Initial Conditions for this high-resolution simulation should be described because they are very important for stratospheric evaluation. The short duration of this simulation should not be an issue in the upper stratosphere, because the ozone lifetime is shorter than in the middle and lower stratosphere.

**Language improvements**

Original text is copied in *italics*, suggestions for corrections are typed in **bold**.

- Line 64: "*present-day atmospheric*  **composition** *and…*"
- Line 78: "*BCC-GEOS-Chem v1.0*  **resulted from** *the integration of…*"

- Line 144: "*Functional advancements that enable feedbacks from aerosol and greenhouse gases to*  **dynamics are** *introduced separately in Section 2.4.*"

- Lines 196-198: "*…future*  **updates** *of GEOS-Chem and HEMCO in the BCC-GEOS-Chem model would only require downloading and substituting the corresponding source code,*  **avoiding** *significant modifications to the source code* **of the other BCC-AGCM modules**."

- Line 210: "*…further modularized HEMCO*  **to enable** *coupling to the Community Earth System Model (CESM)…*"

- Line 217: "*…in the context of rapid* *updates…*"

- Line 222: "*…that* **are** *already implemented in HEMCO* **and** *driven by…*"

- Line 250: "Details of aerosol chemistry and its interaction with gas-phase chemistry  **are** provided in Lu et al. (2020)"

- Line 252: "*... and* **the** *ISORROPIA II (Fountoukis and Nenes, 2007) thermodynamic module.*"

- Lines 258-259: "*The scheme requires* **the** *input of geography data…*"

- Lines 276-278: "*…instead of using* **the** *convection module in BCC-AGCM-Chem described by Wu et al. (2020).*  *the BCC-AGCM-Chem scheme for wet deposition is hard-coded*  **,** *incompatibi***le** *with the updated chemical species***,** **and**  *lacks the*  *scavenging* **of** *water-soluble species in convective updrafts....*"

- Lines 283-284: " **This** *incorporation*  **relies on** *the radiative transfer module* **already**  *implemented in BCC-ESM1 (Wu et al., 2020).*"

- Line 285: "*…* **and** *incorporati***es** *explicit parameterization*s *for* **the** *major absorbers...*"

- Line 293: "*The aerosol direct effect*  *the mass mixing ratios of bulk aerosols, which are prognostic variables in* **the** *GEOS-Chem chemical module.*"

- Line 301: "*The indirect effects of aerosols involve their role*  *as cloud condensation nuclei and*  **their** *influence on…*"

- Line 306: "*...diagnosed from* **the** *GEOS-Chem chemical module.*"

- Lines 310-311 (suggestion to remove words stating the obvious): "*…where $\beta$ is a scaling factor*  **,** $\rho w$ *is the water density,* **and** *LWC is the cloud liquid water content*  **.** **B**oth $\rho w$ **and** LWC *are diagnosed by BCC-AGCM.*"

- Lines 313-317: "*The detailed treatment of* **the** *aerosols feedback on precipitation can be found in Wu et al (2020). The current approach for describing* **the** *aerosol feedback usi***es** *a bulk-mass representation of aerosols in BCC-GEOS-Chem v2.0.* **This approach** *is similar to that used in the majority of CMIP5 models,*  **since** *only two of*  **these models** *include online size-resolved aerosol microphysics (Kodros and Pierce, 2017). This bulk-mass representation of aerosols*  *is computationally efficient, but*  *do*e**s** *not consider...*"

- Lines 320-322: "*…microphysics*  **scheme** *(Kodros and Pierce, 2017) from GEOS-Chem to more accurately simulated size-dependent aerosol chemistry and microphysics. Such developments will be available through BCC-GC-HEMCO interface once these two schemes become compatible with the GEOS-Chem column structure.*" Consider a re-formulation starting

e.g. with "**These two schemes are available in GEOS-Chem and simulate more accurately the size dependency of aerosol chemistry and microphysics, but are not yet integrated in BCC-GEOS-Chem v2.0 because…**"

- Line 334: "*…under* **the** *CMIP6 framework,…*"

- Lines 335: "**The** *model results in 2012-2014 are used for…*"

**Additional bibliographical references**

Froidevaux, L., Kinnison, D. E., Wang, R., Anderson, J., and Fuller, R. A.: Evaluation of CESM1 (WACCM) free-running and specified dynamics atmospheric composition simulations using global multispecies satellite data records, Atmos. Chem. Phys., 19, 4783–4821, https://doi.org/10.5194/acp-19-4783-2019, 2019.

Zhang, J., et al.: BCC-ESM1 model datasets for the CMIP6 Aerosol Chemistry Model Intercomparison Project (AerChemMIP). *Adv. Atmos. Sci.* , 38(2), 317−328, https://doi.org/10.1007/s00376-020-0151-2, 2021.